# Microfluidic Devices and Microfluidics-Integrated Electrochemical and Optical (Bio)Sensors for Pollution Analysis: A Review

**Badriyah Alhalaili [1], Ileana Nicoleta Popescu [2],*, Carmen Otilia Rusanescu [3] and Ruxandra Vidu [4,5],***

1 Nanotechnology and Advanced Materials Program, Kuwait Institute for Scientific Research, P.O. Box 24885, Safat 13109, Kuwait

2 Faculty of Materials Engineering and Mechanics, Valahia University of Targoviste, 13 Aleea Sinaia Street, 130004 Targoviste, Romania

3 Faculty of Biotechnical Systems Engineering, University POLITEHNICA of Bucharest, 060042 Bucharest, Romania

4 Faculty of Materials Science and Engineering, University POLITEHNICA of Bucharest, 060042 Bucharest, Romania

5 Department of Electrical and Computer Engineering, University of California Davis, Davis, CA 95616, USA

\* Correspondence: ileana.nicoleta.popescu@valahia.ro (I.N.P.); rvidu@ucdavis.edu (R.V.)

**Abstract:** An overview of the recent research works and trends in the design and fabrication of microfluidic devices and microfluidics-integrated biosensors for pollution analysis and monitoring of environmental contaminants is presented in this paper. In alignment with the tendency in miniaturization and integration into "lab on a chip" devices to reduce the use of reagents, energy, and implicit processing costs, the most common and newest materials used in the fabrication of microfluidic devices and microfluidics-integrated sensors and biosensors, the advantages and disadvantages of materials, fabrication methods, and the detection methods used for microfluidic environmental analysis are synthesized and evaluated.

**Keywords:** microfluidic devices; optical/electrochemical sensors; (nano)biosensors; pollution analysis; environmental contaminants monitoring

## 1. Introduction

Today, human society is facing significant pollution of the environment [1–5] and a massive decrease in natural resources [6–9], leading implicitly to a decrease in the quality of life. The sources of environmental pollution are the result either of natural causes or human activities, such as continuous urbanization and industrialization, excessive exploitation of natural resources, burning of fossil fuels, etc., which affect human health and destroy the balance of the ecosystem. As a result, scientists have been working together to find effective solutions for monitoring and reducing pollution sources by developing advanced materials or exploiting micro/nanodevice fabrication and integration of various processes in clean technologies for environmental sustainability [10–14].

One of these solutions is the use of microfluidic devices and microfluidics-integrated (electrochemical/optical) biosensors for pollution analysis to obtain a quick, accurate, reliable response and rapid diagnosis [9,15–17]. The microfluidic devices allow the integration and miniaturization of an entire laboratory on a very small scale, allowing their integration in a simple and portable system [16], with the advantage of significantly reducing the consumption of reagents, energy, time, and money [15].

Manipulating nano-, pico-, or femtoliter volumes of fluids [15], microchannels serve as electronics, sensors, valves, pipes, and other structures [16,18]. These structures integrated into systems can perform analyses (or other laboratory processes) on a chip called a "lab-on-a-chip" in the range of millimetric dimensions [15,16].

The current trend in sensor technologies is to develop labs-on-chips that allow, for example, the diagnosis of diseases in a very short time, or testing/monitoring (medical diagnostics [19], food, environmental monitoring, etc. [20,21]) in the field (i.e., point-of-need/care) [20,22] outside of central laboratories with devices that are affordable and easy to use by anyone, anywhere, and at any time.

Microfluidic devices—alone or integrated in sensors—have become increasingly important tools for the control of pollution levels in air, water, or soil. One of the important advantages of using advanced materials and/or technologies—such as microfluidic devices integrated in biosensors—is the continuous and real-time monitoring of environmental contaminants such as toxic heavy metal ions, organic contaminants (e.g., phenols/phenolic compounds, pesticides/insecticides), pathogenic microorganisms, or gas pollutants [23,24] for a sustainable environment.

This review paper presents the most common and newest materials used in the fabrication of microfluidic devices integrated in sensors and biosensors, their advantages and disadvantages, and the standard and new detection methods for microfluidic environmental analysis of organic contaminants, pathogenic microorganisms, and toxic heavy metal ions.

## 2. Environmental Pollution: Pollution Types and Potential Solutions for Their Reduction/Sustainable Management

As is known, there are three major types of pollutants that cause degradation of the natural environment, namely, water, soil, and air pollutants. Of the gaseous or air pollutants, the most common are $CO$, $CO_2$, $NO_2$, $SO_2$, $H_2S$, and volatile organic compounds—those that are released directly into the atmosphere and affect both the environment and the health of people and/or animals [25–27].

Domestic or industrial waste pollutes water and soils with heavy metals, hydrocarbons, inorganic and organic solvents, plastics, etc. [28–30]. A first step in order to solve the problems related to pollution is the development of new technologies and economical approaches for the continuous monitoring of pollution sources; removing polluting factors; establishing strategies to protect the atmosphere, continental or maritime waters, and soils; and/or increasing the efficiency of using natural resources in accordance with the actual legislation—for example, implementation of Agenda 2030 for Sustainable Development [3,17].

One of the novel developments in advanced materials and technologies is the use of microfluidic and lab-on-a-chip devices for pollution analysis. Microfluidic devices and microfluidics-integrated sensors represent powerful analytical tools for the real-time and in situ detection of different types of micropollutants present in aquatic systems, with high sensitivity and specificity [30].

The applications of microfluidic devices for the detection of the most common pollutants are presented schematically in Figure 1.

In the following sections, the most common and newest materials used in the design and fabrication of microfluidic devices, microfluidic detection systems, and microfluidics-integrated (bio)sensors for pollution analysis, along with their advantages and disadvantages, are presented. Furthermore, we synthesized and evaluated the old and new microfluidic detection systems for the environmental analysis of heavy metals, phenolic compounds, pathogens, nitrites, nitrates, and ammonia.

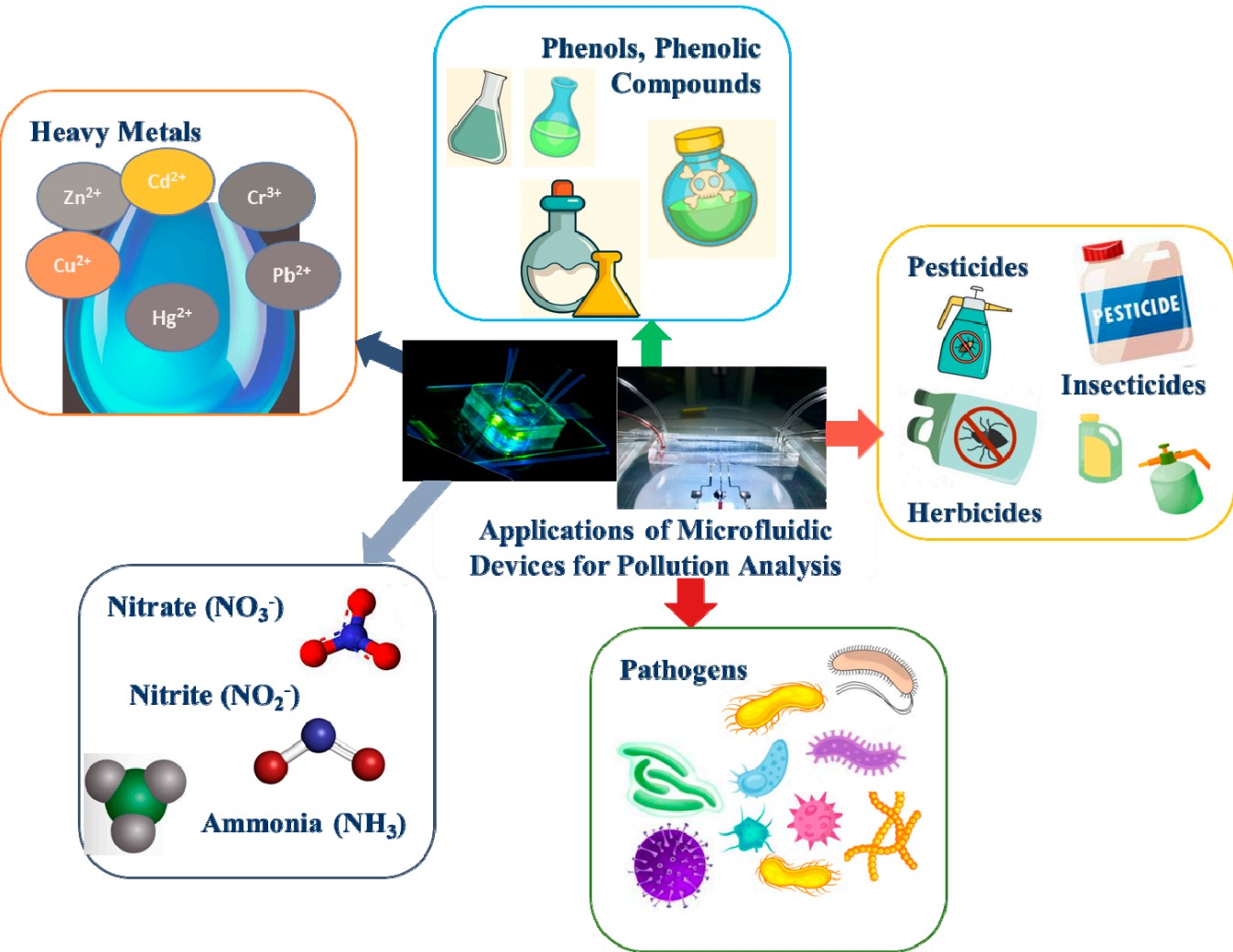

**Figure 1.** The environmental applications of microfluidic devices for the detection of the most common pollutants.

## 3. Design and Fabrication of Microfluidic Devices

*3.1. Component Materials for Microfluidic Devices*

Microfluidic chips are fabricated using the following materials: (a) inorganic materials, such as glass, silicon, ceramic microfluidic chips, and transition metal carbides and/or nitrides; (b) polymeric materials, e.g., polydimethylsiloxane (PDMS) [15] and thermoset polyester (TPE) as elastomers, and polystyrene (PS), polymethylmethacrylate (PMMA), polycarbonate (PC), etc., as thermoplastic polymers and hydrogels, which are relatively novel polymers; and (c) paper-based microfluidic chips [15]. In Table 1, the component materials for microfluidic devices, along with their main characteristics (including advantages and drawbacks) and principal fabrication methods, are briefly presented [15,31–38].

**Table 1.** Material types, characteristics, and fabrication methods for microfluidic chips.

| Material Types | Characteristics | Fabrication Methods |
|---|---|---|
| Silicon (or silicon-based substrates) | (i) Resistant to organic solvents;<br>(ii) Ease in depositing metals;<br>(iii) High thermal conductivity [39];<br>(iv) Stable electroosmotic mobility;<br>(v) High elastic modulus (130 to 180 GPa);<br>(vi) The precise definition of nanoscale channels or pores;<br>(vii) Transparent to infrared [35];<br>(viii) Chemical stability [40].<br>Drawbacks: (ix) Difficulties in handling them (they are hard), making it difficult to make valves and/or pumps, or active microfluidic components in general; (ix) high costs [34] | (a) Wet (chemical) etching [34,35,40–42];<br>(b) Dry etching [43];<br>(c) Powder blasting [33];<br>(d) Micro-hot embossing molding [44];<br>(e) Photolithography [33] |
| Glass (or glass-based substrates) | (i) Optically transparent;<br>(ii) Electrically insulating (amorphous);<br>(iii) Compatible with biological samples;<br>(iv) Not permeable to gas;<br>(v) Has a low (relative) non-specific adsorption.<br>Drawbacks: (vi) Vertical walls are more difficult to etch than Si;<br>(vii) Production is time-consuming and expensive [36] | (a) Wet or dry (chemical) etching [35];<br>(b) Metal or chemical vapor deposition [35];<br>(c) Patterning and cutting [45];<br>(d) Photolithographic patterning [46];<br>(e) Thermal bonding [41];<br>(f) Molding process [47];<br>(g) Powder blasting |
| Al-oxide-based materials | (i) Low-temperature co-fired ceramic (LTCC);<br>(ii) LTCC compared to other technologies allows the integration of heaters, sensors, and electronics (control and measurement electronics, and a light-detection system) into a single module; thus, the measurement system can be simplified;<br>(iii) Thick film materials offer the possibility to fabricate not only the networks of conducting paths in a single package, but also other microsystems, electronic components, and sensors [35].<br>Drawback: No mechanical flexibility | (a) Laminate sheets of Al-oxide-based material are patterned, assembled, and heated at elevated temperatures [48];<br>(b) Electrodes can be deposited onto LTCC using expansion-matched metal pastes [35] |
| Transition metal carbides and/or nitrides and $M_{n+1}X_n$ (MXenes) | (i) High intercalation capacity;<br>(ii) High metallic conductivity [49];<br>(iii) Large surface area;<br>(iv) Good ion-transport properties;<br>(v) Low diffusion barrier;<br>(vi) Biocompatibility;<br>(vii) Hydrophilicity;<br>(viii) Ease of surface functionalization [50];<br>(ix) Higher signal-to-noise ratio in electrochemical sensing [51] | (a) Wet chemical etching [50];<br>(b) Selective etching and exfoliation process [49];<br>(c) Chemical vapor deposition (CVD) growth [52] |
| Polydimethylsiloxane (PDMS) | (i) Optical transparency up to 280 nm;<br>(ii) Ductile (flexible) material;<br>(iii) Elasticity (which can be "adjusted" using crosslinking agents);<br>(iv) Biocompatibility;<br>(v) Sealing capacity of materials such as glass, polystyrene, and PMMA [15];<br>(vi) Does not require a clean room [15];<br>(vii) Permeability to gases (is more permeable to $CO_2$ than to $O_2$ or $N_2$) and water vapor;<br>(viii) High thermal stability up to T = 300 °C;<br>(ix) Cost-effective production at micro scale.<br>Drawbacks: (x) Low shear modulus (e.g., cannot be used at for high-frequency droplet generation at high operating pressure [51];<br>(xi) Swelling in organic solvents;<br>(xiii) Diffusivity [15,32,33] | (a) Device molds made through conventional machining;<br>(b) Device molds made by photolithographic methods [53];<br>(c) Micromolding–casting process (liquid PDMS prepolymer is thermally cured at mild temperatures of 40–80 °C and can be cast at nanometer resolution from photoresist templates [33,53] or other techniques;<br>(d) "Microwire molding" [15,32];<br>(e) Rapid prototyping [54] |
| Thermoset polyester (-TPE) | (i) Insoluble;<br>(ii) Highly resistant to creep;<br>(iii) Optically transparent and absorbs UV light [55];<br>(iv) Inexpensive;<br>(v) Higher elastic modulus (1-100 MPa) than PDMS [56].<br>Drawbacks: (vi) High stiffness (improper for the fabrication of valves);<br>(vii) High cost;<br>(viii) Hydrophobic [35,57] | (a) Polymerization of polyester and styrene through UV or heat [35];<br>(b) Photolithography [58];<br>(c) Replica molding [59] |

**Table 1.** *Cont.*

| Material Types | Characteristics | Fabrication Methods |
|---|---|---|
| Polystyrene (PS) | (i) Optically transparent;<br>(ii) Biocompatible,<br>(iii) Inert;<br>(iv) Rigid,<br>(v) Relatively hard and brittle;<br>(vi) Good electrical properties;<br>(vii) Surface can be easily functionalized;<br>(viii) Excellent gamma radiation resistance [60].<br>Drawbacks: (vii) Difficulties encountered in the thermal bonding step [33];<br>(viii) Hydrophobic (requires chemical modification of styrene PS surface or plasma oxidation to become hydrophilic) [61] | (a) Injection molding [62];<br>(b) Hot embossing [35];<br>(c) Prototyping by UV laser photoablation [38] |
| Polymethylmethacrylate (PMMA or PMMA substrate) | (i) Low cost [63];<br>(ii) Rigid mechanical properties;<br>(iii) Excellent optical transparency;<br>(iv) Compatibility with electrophoresis [37];<br>(v) Biological compatibility [35];<br>(vi) Elastic modulus of 3.3 GPa [35];<br>(vii) Gas impermeability;<br>(viii) Micromachining at 100 °C [35].<br>Drawback: The cost of PMMA substrate per unit area is high [58] | (a) Hot embossing [63];<br>(b) Solvent imprinting;<br>(c) Atmospheric pressure molding [64]; and thermal bonding;<br>(d) Injection molding [62];<br>(e) Laser ablation [65];<br>(f) $CO_2$ laser micromachining [66];<br>(g) Plasma etching [37];<br>(h) Nanoimprinting |
| Polycarbonate (PC) | (i) Good machining properties;<br>(ii) High impact resistance;<br>(iii) Enhanced chemical resistance;<br>(iv) Low water absorptivity (<0.01%);<br>(v) Good electrical insulating properties;<br>(vi) Long-term stability of surface treatments;<br>(vii) Extremely low absorption of impurities;<br>(viii) Low cost;<br>(ix) Durable material;<br>(x) Very high softening temperature (~145 °C) [35].<br>Drawback: (xi) Low transparency in the visible and near-UV spectra | (a) Prototyping by UV laser photoablation [38];<br>(b) Hot embossing [67];<br>(c) $CO_2$ laser machining [68];<br>(d) Injection molding [62] |
| Polyethylene terephthalate (PET) | (i) Resistant to thermal shock in comparison with silicon-based substrates [40];<br>(ii) Inexpensive production [40] | Laser ablation [69] |
| Cyclic olefin copolymer (COC) | (i) Optical transparency in the visible and near-UV spectra; enhanced chemical resistance;<br>(ii) Good electrical insulating properties;<br>(iii) Low water absorptivity (<0.01%);<br>(iv) Extremely low level of impurities;<br>(v) Long-term stability of surface treatments [70] | (a) Micromilling method [71];<br>(b) Photolithography [72,73] |
| Hydrogel | (i) Extremely hydrophilic polymer [74];<br>(ii) High biocompatibility;<br>(iii) High biodegradability.<br>Drawbacks: (iv) Softness of hydrogels;<br>(v) Silk fibroin, collagen, and gelatin have poor processability;<br>(vi) Complex microfluidic networks cannot be created—only simple or 2D ones;<br>(vii) Channel deformation [74] | (a) Photopatterning [75];<br>(b) Injection molding [76];<br>(c) Coaxial extrusion-based 3D printing [77] |
| Paper | (i) Easy to work with;<br>(ii) Can be treated to chemically bind molecules or proteins;<br>(iii) Compatible with biological samples;<br>(iv) Inexpensive material.<br>Drawback: (v) Difficult to distinguish individual channels on the chip [35] | (a) Paper patterning;<br>(b) Photolithography [78];<br>(c) Screen printing [79];<br>(d) Inkjet printing [80];<br>(e) Plasma oxidation;<br>(f) Roll-to roll;<br>(g) Cutting [81,82] and ink-writing [83];<br>(h) Wax printing [83] |

In general, materials used for substrates include glass, ceramics, and silicon. When it is necessary to obtain flexible disposable sensors—for example, in rapid test surgery—plastic sheets made from polyamide, polycarbonate, and polyester can be used.

The physicochemical and mechanical properties of glass/silicon-based microfluidics materials depend on the type of glass, and the most important properties required for them are transparency, solvent compatibility, Young's modulus, rigidity, and operating temperature. In Figure 2, the main characteristics of glass/silicon-based microfluidics are schematically presented.

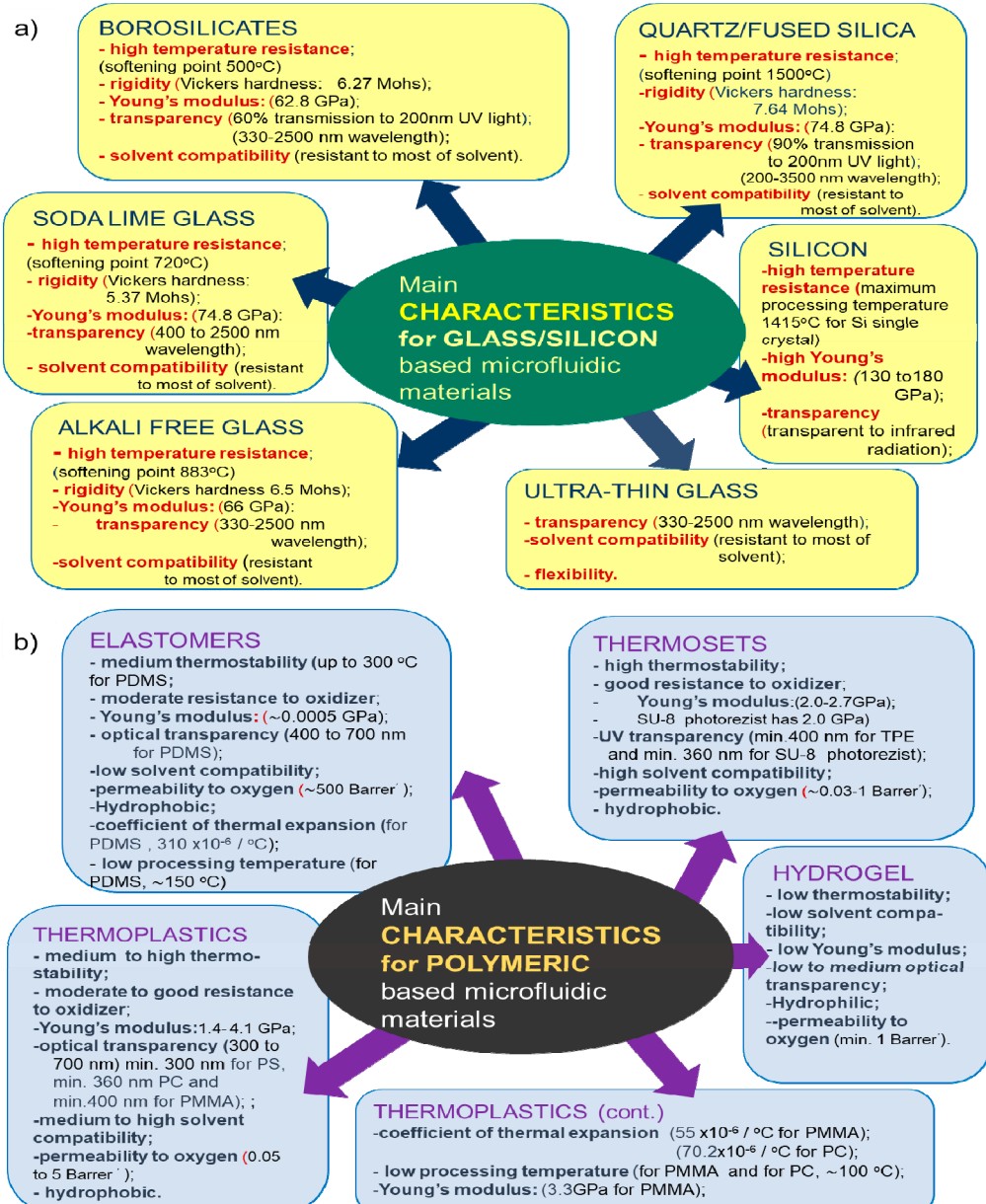

**Figure 2.** The main characteristics of commonly used materials for microfluidic device fabrication: (**a**) glass/silicon-based materials; (**b**) polymer-based materials. PDMS—polydimethylsiloxane, PS—polystyrene, PC—polycarbonate, PMMA—polymethylmethacrylate, TPE—thermoset polyester; $1\,\text{Barrer} = \frac{10^{-10}\text{cm(STP)}\cdot\text{cm}}{\text{cm}^2\cdot\text{s}\cdot\text{cmHg}}$.

The transparency of borosilicate glass, alkali-free glass and ultrathin glass is in the range of 330–2500 nm in wavelength, while for quartz or fused silica the transparency is in the range of 200 to 3500 nm [84]. In contrast with glass, which is optically transparent, silicon is opaque [33].

The highest operating temperature for quartz and fused silica is 1500 °C. Silicon and glass are resistant to most organic solvents, with the following exceptions: glass has no solvent compatibility with HF, and silica has no compatibility with KOH [33,84].

The silicon/glass-based materials for microfluidic device fabrication also have a very stable surface charge, limited 3D channel profile, and the possibility to achieve the smallest channel at the nano level. For instance, the smallest (16 nm deep glass nanochannels) were reported by Pinti et al. [85], who fabricated chemically uniform nanochannel networks with an ultralow aspect ratio in borosilicate glass substrates, designed to perform multiple unit operations on a single chip. For electrodes, any noble metals used for conventional macroscopic electrodes can be adapted [86].

The new ceramic materials used as components of microfluidic devices include transition metal carbides and/or nitrides, and $M_{n+1}X_n$ (MXenes) [49,50], which are characterized by high metallic conductivity, large surface area, good ion-transport properties, a low diffusion barrier, biocompatibility, and ease of surface functionalization [50].

The polymer-based microfluidic materials are the most used materials for the fabrication of microfluidic chips, because of the specific characteristics presented succinctly in Table 1 and Figure 2b, such as optical transparency (PDMS, TPE, PS, COC), flexible materials (PDMS), biocompatibility (PDMS, PMMA, PS), etc. Among them, hydrogels have specific characteristics enabling them to mimic natural mechanical and structural cues for cell adhesion, proliferation, and differentiation [87]. Moreover, hydrogel materials are used to construct complicated and large-scale tissues with high cell density, high metabolic requirements, and intricate architectures [74]. Despite the specific characteristics of hydrogels—such as extreme hydrophilicity [74], high biocompatibility, and high biodegradability—hydrogels are not frequently used as fabrication materials, because maintaining the device's integrity is quite challenging and can limit their use in the long term.

Another important material used in microfluidic devices is paper. Paper represents a highly useful supporting material for developing sensing devices due to its various advantages, such as low cost (200 times less expensive than PET and 1000 times less expensive than glass) [80], ease of printing, high hydroscopic properties, and biodegradability. There are different types of paper used as substrates in the manufacture of microfluidic devices, such as (i) Whatman chromatography paper, characterized by being hydrophilic, reproducible, and homogeneous, with a clean surface, uniform thickness, wicking properties, medium retention, medium flow rate, and biocompatibility; (ii) glossy paper, characterized by being transparent, degradable, and easy to chemically modify; (iii) nitrocellulose (NC) membranes with specific characteristics including a microporous polymeric surface, high binding capacity for biomolecules, combustibility in air, stability, and reproducibility; (iv) paper towels (translucent and permeable); and (v) ITW TechniCloth wipers (composed of cellulose and polyester) [80].

### 3.2. Microchip Fabrication

Microfluidic devices can be fabricated using different techniques that include prototyping techniques, such as replica molding [15,32], rapid prototyping [15,36], soft lithography [15,86,88–90], injection molding [15,37], and hot embossing [15,33,35]. Other fabrication techniques include X-ray lithography [15,62,89,91], photolithography/optical lithography [15,88,92] or photolithography followed by etching and bonding [15,89], and direct fabrication techniques such as laser photoablation or laser micromachining [15,30,37,38,69]. In Table 2, the advantages/disadvantages of different fabrication methods used for microchip fabrication are presented.

**Table 2.** The advantages/disadvantages of different methods used for microchip fabrication.

| Fabrication Methods | Material | Advantages | Disadvantages | Ref. |
|---|---|---|---|---|
| Soft lithography | PDMS | High resolution (down to a few nm); real-time detection; portable; disposable; cost-effective; able to fabricate 3D geometries | Requiring high sample concentration; pattern deformation; vulnerable to defects | [90,93] |
| Hot embossing | PMMA | Cost-effective, precise, and rapid replication of microstructures; mass production | Restricted to thermoplastics; difficult to fabricate complex 3D structures | [94] |
| Injection molding | Thermoplastic polymers | Easy to fabricate complex geometry, fine features, and 3D geometries; low cycle time; mass production; highly automated | Restricted to thermoplastics; high-cost molds; difficult to form large undercut geometries | [62] |
| Laser photoablation | PET | Rapid; large-scale production | Multiple treatment sessions; limited materials | [30,69] |
| Conventional photolithography/opticallithography | Polymers | High wafer throughputs; ideal for microscale features | Usually requires a flat surface to start with; requires chemical post-treatment | [92] |
| Photolithography | PDMS | Portability; cost-effective and high automation; high sensitivity | Low throughput | [95] |
| Electron-beam lithography | SU-8 3010 | Good resolution; can be precisely aligned | Expensive; requires more time to fabricate | [96] |
| X-ray lithography | PMMA | High resolution to fabricate nanopatterns; absorption without spurious scattering; able to produce straight, smooth walls | Difficulties in master fabrication process; time-consuming; high cost | [91] |
| Photolithography and complex pattern | Whatman No.1 chromatography paper, ITW TechniCloth, and Scott hard roll paper towels | Mass production; good stability | Expensive equipment; toxic reagents; fragile when bending | [80] |
| Photolithography or wax printing | SU-8 | Simple; portable; fast; low cost | - | [97,98] |
| Wax printing | Whatman No.1 chromatography, Whatman filter paper, and nitrocellulose (NC) membranes | Simple and fast to fabricate; mass production | Low resolution; not resistant to high temperatures | [99,100] |
| Inkjet printing | Filter paper | Cheap reagents; mass production; compatible with multiple functional inks | Requires an improved ink jet printer; low speed | [101] |
| Inkjet etching | Filter paper | Cheap reagents; prints flexible, foldable channels at 100 cm$^2$ in size | Low resolution; low production; not suitable for complex patterns | [101, 102] |
| Screen printing | Whatman No.1 filter paper | Low cost; mass production; multiple functional inks | Low resolution; different patterns need different printing wire | [79] |
| Nanoimprinting | PMMA | Cost-effective; high sensitivity; high resolution; precise control | Expensive; low throughput | [103] |

## 4. Microfluidic Detection Systems and Microfluidics-Integrated (Bio)Sensors for Pollution Analysis

*4.1. Sensor Types and Their Required Characteristics for the Detection and Monitoring of Environmental Contaminants*

Sensors are devices that can analyze the target analyte quantitatively based on the interaction between the recognition element and the target samples. There is a wide range of sensing devices, classified depending on the detection mode and their measurable properties, such as bio(chemical) [19], electrochemical [30], piezoelectric [104], optical [28], thermal [30,105], magnetic [106], or magneto-optical sensors that provide critical analytical information in many fields [40,107], as illustrated schematically in Figure 3. These types of sensors are able to recognize the analyte of interest on the surface of a signal transducer, depending on the chemical, electrical, optical, magneto-optical, or thermal signal acquisition.

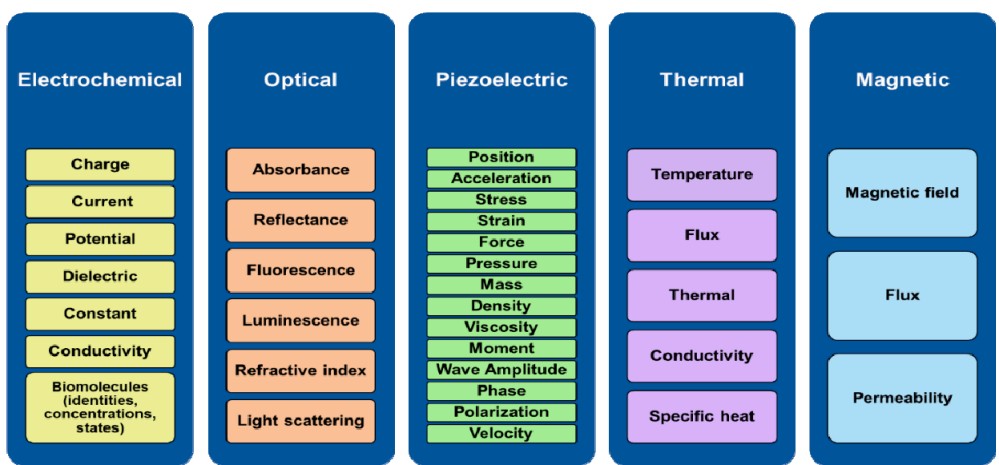

**Figure 3.** Classification of sensors depending on the detection mode and their measurable properties.

Among them, biosensors are analytical devices that consist of (i) a bioreceptor (i.e., a biological recognition component), (ii) a physicochemical transducer to generate a measurable signal, and (iii) an element for signal amplification and processing [108,109]. Biosensors can be classified into nucleic-acid-based biosensors, antigen-based biosensors, or antibody-based biosensors, depending on the biological molecule (i.e., nucleic acids, antigens, or antibodies).

The optical and electrochemical biosensors have been successfully used in biological, chemical, and biomedical analysis, in the detection of biological targets [21,88], in cell culture studies [32], in environmental analysis/monitoring [20,21,110,111], in food analysis [112] and control [110], and in drug discovery and delivery [110].

Optical biosensors consist primarily of (i) a light source, (ii) optical components used to generate and focus the light beam to a modulating agent, respectively, (iii) a modified detection (sensing) head, and (iv) a photodetector [30,113]. Energy, polarization, absorption, fluorescence, light scattering, amplitude, decay time, and/or phase [114] are different parameters that can be used in the optical detection of targets.

An electrochemical (bio)sensor consists of (a) a receptor that recognizes the species to detect it with high specificity and selectivity, and (b) a transducer that translates the event of recognition into a measurable physical (i.e., electrical) signal [115]. The electrochemical devices used as sensors present the most promising advantages in comparison with various classes of elements able to transduce a chemical or biochemical event into a measurable signal, or in comparison with the conventional methods.

Among the most important advantages of the electrochemical devices used as sensors are their flexibility, ability to perform analysis in a short time, low fabrication costs, and ease of implementation and disposability (i.e., easy-to-use sensing devices) due to miniaturization of the electrochemical systems by coupling microfluidics with electrochemical

detection analysis [111]. However, the challenge relates to the fabrication of the miniaturized electrochemical systems due to the thick electrodes that have to be integrated within the microfluidic microelectromechanical systems (MEMSs) and nanoelectromechanical systems (NEMSs) [110]. In both MEMS and NEMS devices, the electrodes used in the electrochemical measurements have dimensions in the micrometric range, in comparison with the traditional electrochemical analysis devices, which are of millimeters in size. The micro- and nanoelectrodes offer the following advantages: measurement of small currents at pico- and nanoampere levels, rapid response to changes in applied potential, low ohmic reduction in electric potential, efficient diffusional mass transport (at microliter sample volumes), and steady-state response to diffusion-controlled potential [21].

Electrochemistry has been and still is used to study the heterogeneous kinetics of electron transfer at the metal–solution interface [21]. Electrochemical phenomena are measured using a three-electrode cell consisting of (1) a working electrode (WE) where redox reactions occur, (2) a counter electrode (CE) that is controlled by the potentiostat to set the potential of WE and the equilibrium current, and (3) a reference electrode (RE) that provides a response to the WE potential to the potentiostat [21,111]. The WE and CE are immersed in the solution being studied, and the RE is often in indirect electrical contact with the help of a conductive salt bridge [21,111]. Recently, miniaturized electrochemical biosensors have shown the advantages of real-time monitoring and label-free detection of biomarkers [116].

The piezoelectric materials used in sensors determine the mechanical resonance of the vibrating crystal at its natural frequency. As the analyte of interested is exposed to the sensing material, a reaction will eventually occur and produce a shift in the frequency that causes a change in the electrical signal. The research of piezoelectric biosensors integrated with microfluidics is quite underdeveloped so far. Possible reasons could include their low sensitivity, poor biocompatibility, and complicated fabrication [17].

Several characteristics of sensors can be obtained to determine the response capability and performance. The optimization of these characteristics is critical to assessing the performance of the sensors.

The main parameters that determine the quality of biosensing are selectivity, sensitivity, and linearity. Figure 4 summarizes the key parameters in the evaluation of biosensors' performance.

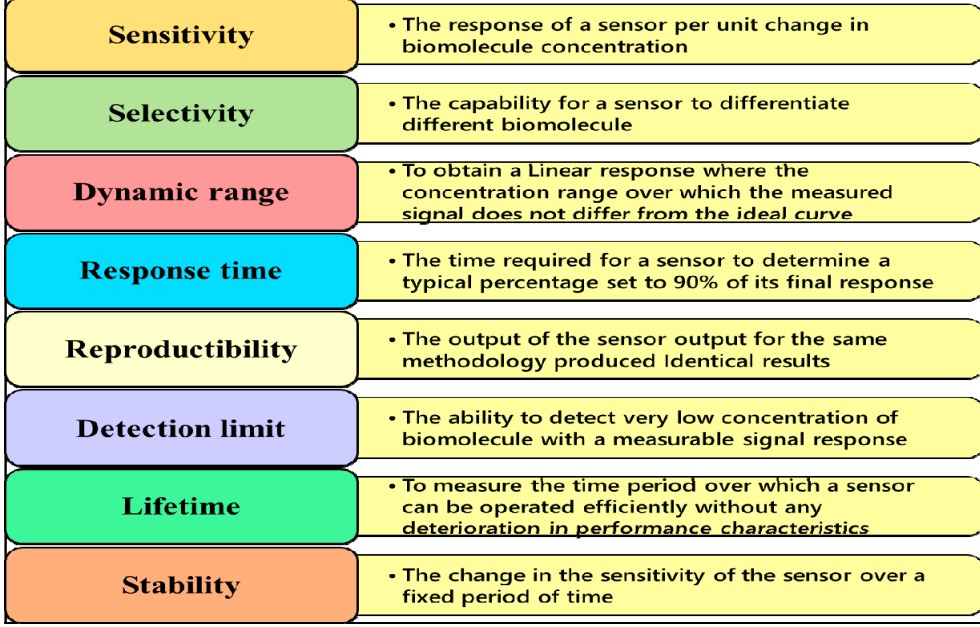

**Figure 4.** Characteristics used for the evaluation of (bio)sensor performance.

Selectivity: Selectivity is the most critical feature of a biosensor (e.g., in case of interaction of an antigen with the antibody) and requires special attention when selecting the suitable bioreceptors [117]. The selection of a proper bioreceptor leads to the detection of a specific bioanalyte in a sample containing other mixtures and chemicals. The interaction model of an antigen with the antibody is one of the best examples to describe the selectivity of bioreceptors. For instance, bioreceptors (e.g., antibodies) are immobilized on the surface of the transducer. When an antibody is exposed to the surface, it will interact only with the antigens, leading to the successful detection selectivity of the target biomolecule and better performance of the sensors or nanobiosensors.

Sensitivity: In various medical and environmental applications, nanobiosensors are needed to detect concentrations of the target analyte in samples as low as nanograms per milliliter (ng/mL) or even femtograms per milliliter (fg/mL) [117]. It is known that portable onsite biosensors, due to the open environment of analysis, affect (decrease) the detection results. Conversely, through biosensors integrated with microfluidic devices, because microfluidics provide a closed and stable biosensing environment, sensitivity is improved and, as a result, the performance of the biosensor is enormously increased [17].

Linearity: In biosensors and nanobiosenors, linearity or linear range (LR) is the feature that measures the change in the range of the nanobiosensor's response to the bioanalyte concentrations linearly with the concentrations. Linearity is related to the resolution of the (bio)sensors and nanosensors, where the detection of the smallest change in the analyte concentration is required to measure the change in the nanobiosensor's response. From the instrumentation point of view, sensor fabrication requires a linear response. Whether a linear or nonlinear response is obtained can be determined based on the objectives of the fabricated devices. Even though the observation of nonlinear responses leads to consistent, repeatable, and predictable results, from the instrumentation point of view, a linear response is highly desirable in the fabrication of sensors. In terms of sensor calibration, the linear region of the input–output values helps to perform the mathematical calibration for the unknown. Therefore, the consistency in the linear variation of the sensor also determines the stability of the device.

### 4.2. Miniaturization and Integration of Electrochemical Sensors in Microfluidic Systems

Typically, a microfluidic MEMS comprises a set of microchannels and microelectrodes, in which the latter are designed based on a model that can control the flow of a selective and sensitive fluid inside the microchannels. Essentially, two separate substrates are required to obtain a functional microfluidic MEMS, where the microchannels that are encompassed in the first substrate are sealed by bonding to the second substrate where the microelectrodes are located [110]. The idea of combining microchannels with electrochemical techniques has its roots in the early days of microfluidics, when electrophoretic separations used microchannels for filtration [21,40,86].

Electrochemical sensors are the most studied sensors [118], and they are typically based on a redox reaction involving the target analyte in the electrolyte at the WE, resulting in variation in the electrical signal [88,118]. When we measure the current and the potential difference between electrodes, the method is called amperometry and potentiometry, respectively. Potentiometry is usually used for ion-selective electrodes' (ISEs) measurements (e.g., pH electrodes, other ISEs). One of the most used electroanalytical methods is voltammetry or voltamperometry. This method is a subclass of amperometry, which measures the current as the applied potential is varied [118]. Cyclic voltammetry is performed by applying an up-and-down linearly varying potential between the WE and the RE, and then plotting the current generated externally from the CE to the WE; the resulting curve is called a cyclic voltammogram or CV [118].

In electrochemical impedance spectroscopy (EIS), an alternating voltage is applied, with a frequency that varies from $10^{-3}$ to $10^5$ Hz, and the current is measured at the same frequency [118]. The results are analyzed on a Nyquist diagram [118], with the imaginary part as a function of the real part [88,118]. EIS is a powerful electrochemical method

that has recently become popular in biosensitivity due to its ability to detect binding events on a transducer's surface. In EIS, a DC potential (EDC) and a small sinusoidal AC perturbation (EAC, B5–10 mV amplitude) are applied between the WE and the RE. The magnitude and the phase angle ($\theta$) of the resulting current ($I$) are recorded as a function of the AC frequency [21]. EIS facilitates the extraction of device-specific parameters from an equivalent circuit model, and these parameters are used to describe the performance of the microelectrodes in a microfluidic channel [119]. Thus, stable Ag/AgCl microelectrodes, manufactured using a combination of photolithographic and electroplating techniques, have been shown to be useful for electrochemical analysis in microfluidic systems [119].

### 4.2.1. Microelectrode Materials Used in Electrochemical Device Sensors

One of the most important factors in designing an electrochemical sensor is the choice of material for the WE. The electrodes must be suitable for the specific application (i.e., chemically resistant to the sample, chemically stable over time, etc.) and have specific characteristics such as sensitivity, selectivity, or long-term stability. The most used materials for electrodes with applications as sensors are (i) carbon-based materials (e.g., screen-printed carbon electrodes, carbon fibers, diamond, etc.) [120]; (ii) metallic electrodes, such as Au [121], Pt [86], Pd [122], Cu [123], Ni [124], Hg/Au amalgams, or Bi; and (iii) semiconductor metal oxides [20,40].

For different electrochemical applications, using activated charcoal, magnesium, or melanin, there has recently a great interest in producing biodegradable and compostable electrodes [20]. Carbon electrodes are used in electrochemical detection because (i) the fouling is minimal [40], (ii) the potential range for organic compounds is larger than that of metal electrodes [40], and (iii) they use low-noise metal microwires (less than 50 µm) as the working electrodes for electrochemical detection using platinum, gold, and copper [121]. Wang et al. [122] demonstrated that a Pd electrode had better detection sensitivity for hydrazine than a carbon electrode in electrochemical sensors. They showed that when the Pd electrode was used, the signal recorded presented sharp peaks and an improved signal-to-noise ratio [40,121]. By using gold electrodes for the detection of phenolic compounds (e.g., chlorophenols, aminophenols), the signal-to-noise ratio was greatly improved, while the peaks became sharper compared to other electrodes. Noble metal electrodes bring another advantage, namely, the electrocatalytic effect [40,121]. By using Cu electrodes, sugars can be detected, while when using Ni as a working electrode, electrocatalytic effects towards aliphatic alcohols and sugars have been demonstrated [40,124].

Electrodes made of Hg/Au amalgams have the required electrochemical properties to detect nitroaromatic explosives [40,125], while electrodes made of Bi show similar electrochemical properties to carbon electrodes, with a similar signal-to-noise response [40,125].

### 4.2.2. Microelectrodes Fabricated for Use in Microfluidic Detection Systems and Microfluidics-Integrated (Bio)Sensors

The development of electrochemical sensors uses certain design criteria, such as (i) miniaturized manufacturing design [111], (ii) sensitivity and selectivity [111], (iii) robustness, (iv) reversibility, (v) speed, (vi) automation, (vii) reliability, (viii) stability, (ix) data acquisition, (x) compound analysis capabilities, (xi) low power consumption, and (xii) overall cost [35,111]. The innovative techniques used for making microfluidic electrochemical devices used as sensors include thick- and thin-film technology (metallization) [111], chemical etching, and photolithography. Using these methods, two-dimensional sensors have been fabricated [88].

The most used methods are wet or dry chemical etching in combination with the pattering of photoresistors. The steps involved in forming thin-film metal electrode patterns by chemical etching are (i) deposition of a metal layer, (ii) spin-coating of a photoresistor and pre-baking, (iii) exposure to UV light and development, (iv) rinsing of the photoresistor, (v) etching of the metal layer, and (vi) removal of the photoresistor [86,119].

The steps involved in forming thin-film metal electrode patterns by lift-off techniques are (i) spin-coating of a photoresistor and pre-baking, (ii) exposure to UV light, (iii) soaking in an aromatic solvent and development, (iv) rinsing of the photoresistor after baking, (v) deposition of a metal layer, and (vi) removal of the photoresistor. The lift-off technique has been used for patterns of noble metals (i.e., Pt and Ir) [86].

The fabrication of pumps, valves, and other microfluidic components is contingent on bulk micromachining to create microscopic 3D structures in a silicon substrate [86,88]. For 3D structures, some researchers [89,126] have applied field-assisted bonding or anodic bonding techniques that consist of sealing glass–metal, glass–semiconductor, and glass–glass systems [89,126]. They demonstrated that field-assisted glass sealing offers a simple and rapid method of making reliable, strong hermetic bonds at low temperatures [89,126]. The copper electrodes for conductivity detection can be fabricated on a printed circuit board attached to a PDMS−glass device; Pd electrodes can be fabricated on glass plates before bonding with PDMS for amperometric detection [88,123].

Of the many techniques to fabricate microelectrodes for use as sensing devices, electrochemical deposition has recently been progressively used for generating thick electrodes integrated within microfluidic MEMSs [119,127]. In addition, electrochemical deposition offers a simple procedure for the manufacture of microelectrodes that are made of different types of metals [119,127]. The electrodeposition of nanocrystalline metals and alloys has been investigated by many researchers [127].

From a nanostructure point of view, electrochemical deposition is used in order to obtain laminar metal coatings and freestanding foils, in a single bath or between two baths, by the alternating movement of the growing electrode (an alternative sequence of two different metals) [127]. It has been observed that due to the dynamic characteristics of the electrokinetic process, spontaneous formation of multilayers often occurs via electrodeposition of different nanometric materials, such as Fe-Ni, Zn-Ni, Cu-Sb, or Au-Cu [127]. Gold electrode bases for amperometric biosensors were first prepared on polycarbonate sheets using a photodirected selective electroless gold plating technique [121,128].

Wang et al. [128] prepared a micro-gold-film electrode based on a polycarbonate (PC) coating sheet with a photodirected electroless plating technique. This developed micro-flow-injection biosensor system with PC could successfully be applied for the determination of glucose content in pharmaceutical injections [128]. For environmental monitoring applications, Wang et al. [129] used Si-based techniques to create an electrochemical sensor for the detection of trace metals in natural waters, and achieved remarkable sensitivity (detection of trace Ni and U required only 5 and 20 min, respectively). In their experiments [129], the integrated membrane/electrochemical sampling sensor pursued trace monitoring of uranium and nickel using propyl gallate (PG) and dimethylglyoxime (DMG) as chelating agents. These tests established adsorptive stripping protocols for trace uranium and nickel based on complexation with PG and DMG. Experimental variables including reagent delivery rate and ligand concentration were used to characterize and test the experimental stripping probe. Despite internal dilution, the renewable-flow probe resulted in extremely low detection limits, such as 0.9 μg/L ($1.5 \times 10^{-8}$ M) for nickel and 10 μg/L ($4.2 \times 10^{-8}$ M) for uranium [111,129].

### 4.3. Miniaturization and Integration of Optical Sensors in Microfluidic Systems

The microfluidics integrated in optical sensors are also known as optofluidics. By integrating the optical sensors in a microfluidic system, sample processing and biosensing reactions are performed in a closed and relatively stable environment that allows for fast, high-efficiency, contactless analysis under a well-controlled microenvironment. Other advantages include a low detection limit, versatility, being label-free and non-destructive, and their ability to detect a wide variety of analytes or multiple analytes at the same time with fast signal monitoring and analysis [16,30,71]. Moreover, the simple design of optofluidic systems allows for reducing the cost of the device fabrication as well as precise quantification and detection of different environmental pollutants, including heavy

metal ions, pesticide residues in agricultural foods, herbicides, food allergens, phenolic compounds, pathogens, etc. [130–132].

*4.4. Microfluidic Detection Systems for Pollution Analysis*

Microfluidics can be coupled with a multitude of detection devices for optical detection, electrochemical detection, mass spectrometry, etc. In the case of optical detection, the most common methods for microfluidics are (a) absorbance-based detection, such as colorimetry [71,133,134]; (b) fluorescence detection [135,136]; (c) chemiluminescence detection [137] or bioluminescence [138]; (d) surface plasmon resonance (SPR), with or without fiber optics [139]; and (e) laser-induced fluorescence (LIF) [140].

Colorimetric and fluorimetric detection schemes are well suited for the detection of environmental contaminants in less accessible and remote areas. These methods require only simple equipment, such as a light-emitting diode for excitation used in conjunction with a photomultiplier tube or even a smartphone camera for detection [141]. In addition, fluorescence detection is widely used due to its high selectivity and sensitivity [142].

Due to its instrumental simplicity, availability, flexibility, rapid analysis with high accuracy, low manufacturing costs, and facile implementation, microfluidic devices coupled with electrochemical detection are more advantageous compared to traditional electrochemical detection systems [100]. The main electrochemical detection methods for microfluidics applied for the detection and monitoring of environmental contaminants are (a) (chrono)amperometry [143]; (b) voltammetry, such as square-wave anodic stripping voltammetry (SWASV) [72,73], differential pulse anodic stripping voltammetry (DPASV) [144], cyclic voltammetry (CV) [30], or linear sweep voltammetry (LSV) [145]; (c) conductometry [30,146]; (d) potentiometry [147]; and (e) electrochemical impedance spectroscopy [148].

The voltammetric detection implemented in microfluidic devices, compared to stationary analysis, is associated with improved detection limits, where the faradic current increases due to the increased transport rate of the analyte to the electrode surface [147] for microfluidic applications.

Contactless conductivity is one the most important techniques to detect inorganic or small organic ions in electrophoresis. It is preferred due to the electrodes' fouling, bubble formation due to water electrolysis, and interference with high voltages used to drive electroosmotic flow [149,150]. Conductivity detection can be achieved either by a direct contact of the mixture with the sensing parts or by a contactless method where the sensing electrodes are not attached directly to the measured mixture. This process requires a detector cell as a basic part of the electronic circuitry. To evaluate the performance of the contactless conductivity detection, two major issues need to be addressed: the noise analysis, and the detector's sensitivity.

4.4.1. Microfluidic Detection Systems for Heavy Metals

Over time, researchers have been concerned with the detection and monitoring of heavy metal ions using different types of microfluidic systems or microfluidic sensors, which allow continuous and on-site measurements of heavy metals.

Several optical methods are widely used to identify and quantify heavy metals, including colorimetry [30], surface plasmon resonance, [71,139], fluorescence [30], and chemi/bio luminescence [98,137,138]. Polymer-based optical microfluidic chips for the analysis of heavy metal ions can be made from polymethylmethacrylate (PMMA), cyclic olefin copolymer (COC)—an amorphous polymer—PDMS [142], or polytetrafluoroethylene (PTFE)/perfluoroalkoxy alkane (PFA) tubes [71]. Figure 5 shows the design and construction of a microfluidic platform with COC support (Figure 5a) and a microfluidic chip molded in PDMS and fixed on glass substrate with connected fibers and tubing for the continuous monitoring of Hg(II) (Figure 5b).

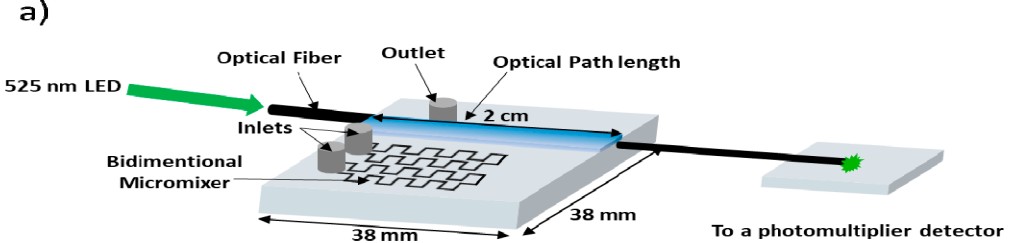

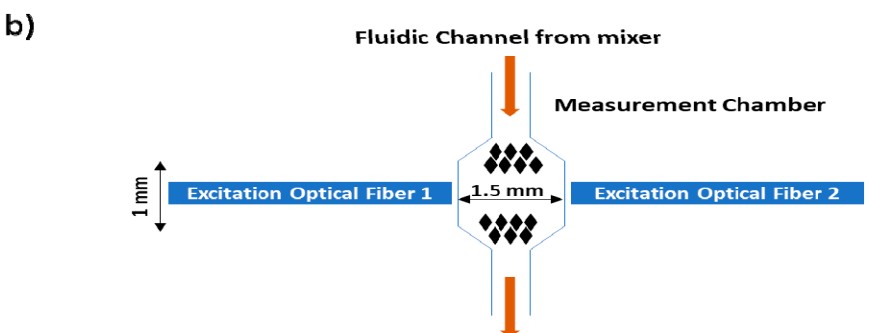

**Figure 5.** The construction of polymer-based optical microfluidic chips for the analysis of heavy metal ions: (**a**) schematic representation of the microfluidic platform with cyclic olefin copolymer support for the continuous monitoring of Hg(II); (**b**) schematic representation of the SU8 mold of the optical chamber.

The detection method of the microfluidic optical system with gold nanoparticles developed by Gomez and collaborators [71] is based on the selective recognition of mercury by a thiourea derivative specifically designed and synthesized for the continuous monitoring of Hg(II). The results obtained using this optofluidic system showed improved analytical characteristics compared to the batch experiments, such as a lower detection limit (11 ppb), higher sensitivity, and faster analysis time, all via an easy, automatic, and low-cost procedure [71]. Mohan et al. [139] also reported the design, fabrication, and characterization of an optical fiber sensor by cascading two channels in a single fiber-optic probe using the SPR technique and ion-imprinted nanoparticles for the simultaneous determination of lead (Pb) and copper (Cu) ions in aqueous samples. The sensing of Pb(II) and Cu(II) ions is based on the interaction of ions with corresponding ion-imprinted nanoparticles. When the solution of the metal ions comes near the ion-imprinted nanoparticle layer, metal ions bind non-covalently with the corresponding complementary binding sites and cause a change in the effective refractive index of the sensing layer (i.e., ion-imprinted nanoparticle layer). The change in the effective refractive index causes a shift in the peak absorbance wavelength of the recorded spectrum. The experimental results showed that the detection limits of both channels were the lowest in comparison with other studies reported in the literature on sensing Pb(II) and Cu(II) ions.

A rapid, eco-friendly, and affordable method for detecting arsenic in water samples was reported by Chauhan et al. [133]. Lace et al. [151] optimized a colorimetric method based on leucomalachite green dye for its integration into a microfluidic detection system. This method can be applied for monitoring wastewater as well as for the detection of arsenic in areas with particularly high arsenic levels.

Table 3 presents a systematic overview of the microfluidic system types, detection methods, fabrication of chips, and specific characteristics of the performance of the optical sensors for different analytes, such as Cr(III) and Cr(IV), Ni(II), Cu(II), Hg(II), Pb(II), Cd(II), and Fe(II).

**Table 3.** Optical microfluidic detection methods for various heavy metal ions.

| Samples | Device Substrate (or Components) | Detection Method (and/or Mechanism) | Fabrication Method | Analyte (Target) | Limit of Detection (LOD) Linear Range (LR) | Ref. |
|---|---|---|---|---|---|---|
| Water sample | Chromatography no. 1 paper | Colorimetry | Patterned paper | Cr(VI) Ni(II) Cu(II) | LOD for Cr(VI): 0.5 mg/L LOD for Ni: 0.5 mg/L LOD for Cu(II): 0.8 mg/L | [152] |
| Sample solution with the addition of nanoparticles (PtNP) | Glass-fiber paper | Colorimetry | Printing technique | Hg(II) | LOD: 0.01 µM | [153] |
| Synthetic samples containing Hg and aqueous NaOH solution (used to extract dithizone from dithizone–CCl4 solution) and then used as a chromogenic reagent | Filter paper | Distance-based colorimetry | Printing technique | Hg(II) | LOD: 0.93 µg/mL | [154] |
| Water sample; sample solution of arsenic prepared in lemon juice | Filter paper | Colorimetric microdetection | Simple pattern-plotting method | As(III) | LOD: 0.01 mg /L | [133] |
| Environmental Samples from (i) Bog Lake; (ii) Killeshin water reservoir; (iii) Laois groundwater; (iv) Barrow Carlow River | – | Colorimetry (absorbance principle) | – | As(III) | LOD: 0.19 mg/L LR: 0.07–3 mg/L | [151] |
| Natural water samples at the sub-ppm range | Paper-based device | Miniaturized chemiluminescence | Wax printing of microfluidic paper-based analytical device (µPAD) | Cr(III) | LOD: 0.02 ppm LR: 0.05–1.00 ppm | [98] |
| Seawater | Polymethylmethacrylate (PMMA) | Colorimetry (absorbance principle) | Micromilling in PMMA of microchannels | Fe(II) Mn(II) | LOD for Fe(II): 27 nM LOD for Mn(II): 28 nM LR for Fe(II): 27–200 nM LR for Mn(II): 0.028–6 µM | [134] |
| Lyophilized (prepared with bacterial luciferase and NAD(P)H:FMN-oxidoreductase) and mixed with aqueous starch suspension | Polymethylmethacrylate (PMMA) | Bioluminescence | Micromilling method | Cu(II) | LOD: 3 µM | [138] |
| Environmental water samples | Cyclic olefin copolymer—an amorphous polymer | Surface plasmon resonance | Micromilling method | Hg(II) | LOD: 11 µg/L LR: 11–100 µg/L | [71] |

**Table 3.** *Cont.*

| Samples | Device Substrate (or Components) | Detection Method (and/or Mechanism) | Fabrication Method | Analyte (Target) | Limit of Detection (LOD) Linear Range (LR) | Ref. |
|---|---|---|---|---|---|---|
| Aqueous samples with mixed concentrations of Pb(II) and Cu(II) ions | Plastic-clad silica (PCS) fiber | Fiber optics + surface plasmon resonance | Coating by thermal evaporation of thick copper and silver film over unclad cores of both channels (I and II); dip-coating of non-imprinted (NIP) nanoparticles over the films; | Cu(II) Pb(II) | LOD for Cu(II): $8.18 \times 10^{-10}$ g/L LOD for Pb(II): $4.06 \times 10^{-12}$ g/L LR: 4.06–1000 μg/L | [139] |
| Aqueous sample solution and aqueous M1 suspension | Polytetrafluoroethylene (PTFE) /perfluoroalkoxy alkane (PFA) tubes | Fluorescence | – | Hg(II) | LOD: 0.02 μg/L LR: 0.02–200 μg/L | [155] |
| Aqueous samples, sewage waters | PDMS/glass | Fluorescence | – | Cd(II) | LOD: 0.45 μg/L LR: 1.12–22.40 μg/L | [137] |
| Natural water | Glass plates | Chemiluminescence + air sampling | Photolithography and wet etching | Fe(II) | LOD: $3 \times 10^{-7}$ mol/L LR: $1 \times 10^{-6}$ to $5 \times 10^{-5}$ mol/L | [137] |
| Diluted stock solution of Fe(II) with demineralized water | Glass | Optical detection (absorbance principle) | Photolithographic and wet-etching techniques; photoresistant coating | Fe(II) | LOD: 1 μM LR: 1–100 μM | [156] |
| Water samples containing certain concentrations of Pb | PDMS substrate | Fluorescence | Molded the channels in PDMS | Pb(II) | LOD: 5 ppb | [142] |

There are many types of microfluidic systems for electrochemical detection, including paper-based microfluidic systems (Whatman paper substrates with different types of electrodes incorporated, e.g., boron-doped diamond paste electrodes) [100], graphite as the WE [30,157–159], polymer-based electrodes such as COC with silver and bismuth as working electrodes, PMMA substrates with boron-doped diamond electrodes [160], gold thin films [161], or PDMS/glass substrates and Au, Pt, etc., as WEs [30].

For instance, Jung et al. [73] made a reusable polymer lab-on-a-chip sensor with a microfabricated silver working electrode for detection using SWASV measurement of lead ions in nature. One of the advantages of this polymeric COC-based microfluidic sensor is its reusability. Thus, after 43 consecutive measurements, it was observed that the peak potentials were stable and the dynamic response was in the range of concentrations from 1 ppb to 1000 ppb [73]. The silver WE was microfabricated and replaced, for instance, the conventional mercury and bismuth electrodes used for SWASV detection by Zou et al. [72]. Gutiérrez-Capitán et al. [161] detected copper ions in different electroactive samples of pollutants with a PMMA-based microfluidic system and Au thin-film electrodes (Figure 6a,b). The copper ions were detected using anodic stripping chronoamperometry (AS-CA) (deposition at −0.40 V and stripping at +0.05 V) with a compact flow system including two electrochemical transducers integrated into a miniaturized cell. Figures 6 and 7 present the components and the construction of electrochemical microfluidic chips. Table 4 presents the microfluidic system types, detection methods, fabrication methods of chips and working electrodes, and specific characteristics of the performance of the sensors.

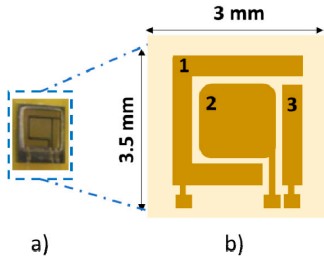

**Figure 6.** The construction of polymer-based electrochemical microfluidic chips for the analysis of environmental pollutants (including heavy metal ions): (**a**) photo of the actual compact electrochemical flow system with PMMA substrate. (**b**) Schematic illustration of the Au thin-film electrodes, where 1—counter electrode (CE), 2—working electrode (WE), and 3—pseudo-reference electrodes.

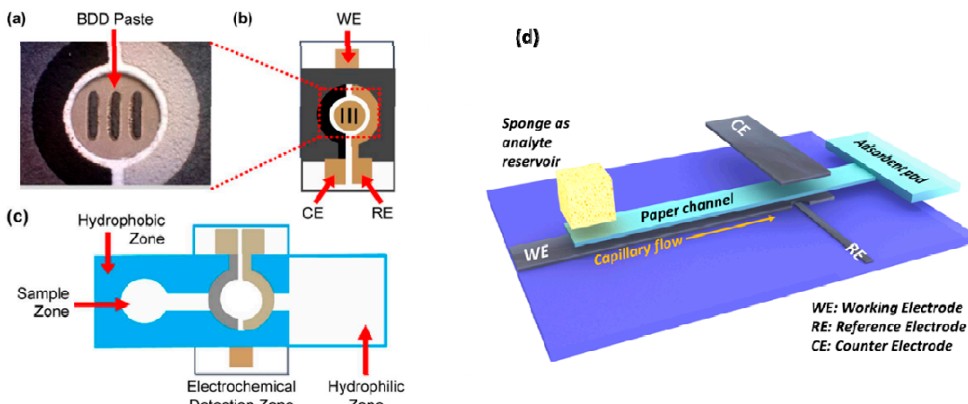

**Figure 7.** The construction of paper based electrochemical microfluidic chips for the analysis of environmental pollutants (including heavy metal ions): (**a**) and (**b**) Boron Doped Diamond Paste Electrodes (BDDPEs) design, (**c**) Schematic illustration of the µPAD (Whatman grade 1 chromatography paper substrate with incorporated BDDPE), for the measurement of Pb and Cd Reprinted with permission from Ref. [100]. Copyright (2017) American Chemical Society; (**d**) Illustration of the microfluidic device based on paper (light blue) with graphite foil WE adapted after Ref. [157] is licensed CC BY 4.0.

**Table 4.** Electrochemical microfluidic detection methods for heavy metals.

| Samples | Device Substrate | Working Electrode (WE) Type | Detection Method | Fabrication Method | Analyte (Target) | Limit of Detection (LOD) Linear Range (LR) and/or Sensitivity | Ref. |
|---|---|---|---|---|---|---|---|
| Real samples of gas-dissolved salty soda water and groundwater with physical contamination | Whatman filter paper | Carbon | Square-wave anodic stripping voltammetry (SWASV) | Screen-printed carbon electrodes (SPCE) on Whatman filter paper | Pb(II) Cd(II) | LOD for Pb(II): 2 ppb LOD for Cd(II): 2.3 ppb LR for Pb(II) and Cd(II): 2–100 ppb | [158] |
| Rice flourdissolved in methanol–water | Whatman filter paper | Boron-doped diamond (BDD) | Square-wave anodic stripping voltammetry (SWASV) | Electrodeposition of gold nano-particles on boron-doped diamond (AuNP/BDD) electrode | As(III) and As(V) | LOD: 0.02 µg/L LR: 0.1–1.5 µg/L | [159] |
| Aqueous solutions | Whatman grade 1 chromatography paper or polyester–cellulose blend paper | Bismuth plated on carbon | Square-wave anodic stripping voltammetry (SWASV) | Photolithography or wax-printing of microfluidic channels; screen-printed electrodes | Pb(II) | LOD: 1 ppb LR: 5−100 ppb Sensitivity: 0.17 µA (µg/L)$^{-1}$ | [90] |
| Aqueous samples (heavy metal stock solutions); mud-spiked samples | Whatman filter paper grade 1 | Graphite | Square-wave voltammetry (SWV) | Wax-printing of microfluidic channels; screen-printing of electrodes | Cd(II) and Pb( II) | LOD for Cd(II): 11 ppb; LOD for Pb(II): 7 ppb LR for Cd(II) and Pb(II): 10−100 ppb Sensitivity for Cd(II): 0.015 µA (µg/L)$^{-1}$ Sensitivity for Pb(II): 0.0025 µA (µg/L)$^{-1}$ | [162] |
| Standard solutions of Cd(II) and Pb(II) | Whatman grade 1 chromatography paper | Boron-doped diamond paste electrodes (BDDPEs) | Square-wave anodic stripping voltammetry (SWASV) | Print wax patterns on microfluidic paper; stencil printed of an electrode with a mixture of BDD powder and mineral oil | Cd(II) and Pb(II) | LOD for Cd(II): 25 µg/L LR for Cd(II): 25–200 µg/L LOD for Pb(II): 1 µg/L LR for Pb(II): 1–200 µg/L Sensitivity of Cd(II): 0.218 µA µM$^{-1}$ Sensitivity of Pb(II): 0.305 µA µM$^{-1}$ | [100] |
| Environmental and biological samples | Cyclic olefin copolymer (COC) | Bismuth | Square-wave anodic stripping voltammetry (SWASV) | Photolithography of COC screen-printed electrode (SPE) | Pb(II) Cd(II) | LOD for Pb(II): 8 ppb; LOD for Cd(II): 9.3 ppb LR for Cd(II): 28−280 ppb LR for Pb( II): 25−400 ppb Sensitivity for Cd(II): 0.065 µA (µg/L)$^{-1}$ Sensitivity for Pb(II): 0.0022 µA (µg/L)$^{-1}$ | [72] |
| Deionized (DI) water for experiments; sample solution (with HNO$_3$ and KCl) for electrolyte; silver electroplating solution for Ag electroplating | Cyclic olefin copolymer (COC) | Silver | Square-wave anodic stripping voltammetry (SWASV) | Spin-coated S1818-positive photoresistor patterned on a COC substrate by a photolithographictechnique; microfabricated silver electrodes | Pb(II) | LOD: 0.55 ppb LR: 1−1000 ppb Sensitivity: 0.028 µA (µg/L)$^{-1}$ | [73] |
| Sample solution containing lead ions | Polymethylmethacrylate (PMMA) | Boron-doped diamond electrode | Square-wave anodic stripping voltammetry (SWASV) | Microelectrodialysercombined with boron-doped diamond electrode | Pb(II) | LOD: 4 µg/L LR: 20–100 µg/L Sensitivity of 15.5 nA L µ/g | [160] |

**Table 4.** *Cont.*

| Samples | Device Substrate | Working Electrode (WE) Type | Detection Method | Fabrication Method | Analyte (Target) | Limit of Detection (LOD) Linear Range (LR) and/or Sensitivity | Ref. |
|---|---|---|---|---|---|---|---|
| Different electroactive pollutants | Polymethylmethacrylate (PMMA) | Gold thin film | Anodic stripping chronoamperometry (AS-CA) | Microfabrication techniques (micromilling in PMMA of microfluidic channels; photolithography of gold thin-film electrodes) | Cu(II) | LOD: <0.3 µM | [161] |
| Water solution containing heavy metal ions | Photosensitive resin | Screen-printed electrode (SPE) modified with $Mn_2O_3$ | Differential-pulse anodic stripping voltammetry (DPASV) | Stereolithographyappearance (SLA) for 3D-printed microfluidic device (prototyping); microporous screen-printed electrode modified with $Mn_2O_3$ | Cd(II) and Pb(II) | LOD for Cd(II): 0.5 µg/L LR for Cd(II): 0.5 to 8 µg/L LOD for Pb(II): 0.2 µg/L LR for Pb(II): 10 to 100 µg/L | [144] |
| Mixture of heavy metal ions | PDMS/glass | Gold | Capillary electrophoresis with contactless detection (CCD) | Spin-coated PDMS membrane on a glass substrate; patterned electrodes in an antiparallel configuration | Heavy metal ions | LOD: 0.4 µM | [149] |
| Sample solution containing mercury ions | PDMS/glass | Screen-printed electrode coupled with sodium-dodecyl-sulfate-doped polyaniline (PANi–SDS | Cyclic voltammetry (CV) techniques and square-wave voltammetry (SWV) | Replica-molding process for PDMS channel; screen-printed electrode (SPE) | Hg(II) | LOD: 2.4 nM LR: 6 nM to 35 nM | [163] |
| Seawater | PDMS/glass | Platinum | Linear sweep voltammetry (LSV) | Soft lithography of PDMS; patterning of electrodes on glass slides; platinum electrodeposition | Pb(II) Cd(II) | LOD for Pb(II): 150 ppb LOD for Cd(II): 340 ppb | [145] |
| Aqueous analyte | Paper substrate | Modifier-free electrodes; graphite foil | Square-wave voltammetry (SWV) | Cutting, stacking | Cd(II) and Pb(II) | LOD for Cd(II): 1.2 µg/L LR for Cd(II): 5–500 µg/L LOD for Pb(II): 1.8 µg/L LR for Pb(II): 5–100 µg/L Sensitivity for Cd(II) and Pb(II): 0.101 µA $(µg/L)^{-1}$ | [157] |
| Lake water and human serum samples | 3D paper-based | Gold nanoparticles (NPs) aggregates and C nanocrystals capped silica NPs conjugated with DNA strands | Electrochemiluminescence (ECL) | Wax-printing and screen-printing | Pb(II) and Hg(II) | LOD for Pb(II): 10 pM LOD for Hg(II): 0.2 nM LR for Pb(II): 30 nM–1 µM LR for Hg(II): 0.5 nM–1 µM | [164] |

### 4.4.2. Microfluidic Detection Systems for Phenols or Phenolic Compounds

Environmental water from natural sources (e.g., seawater, water from lakes, rivers, groundwater, etc.) can be contaminated with various pollutants (see Figure 1), including phenols or phenolic compounds. Detection of toxic substances in water bodies is an important issue in environmental monitoring.

Phenolic compounds are toxic substances and are among the 129 most polluting and most harmful pollutants to human health and the environment controlled and identified by the US Environmental Protection Agency [165].

Phenols and phenolic waste can originate from wastewater discharged by dyes, pesticides, and enterprises—especially petrochemical enterprises [166]—or can be generated during the production of synthetic polymers, such as phenolic resins resulting from the use of coking coal in oil refineries. Another source of phenolic waste is pesticides with phenolic skeletons; these pesticides, through degradation, release phenolic compounds, which contaminate the environment. For instance, chlorophenols are commonly used as pesticides, herbicides, and disinfectants in modern societies, and can also be produced through chlorination of phenols during water disinfection processes [167], etc.

The most common phenolic compounds are phenol, bisphenol A, catechol, cresol, dopamine, epinephrine, 2,4-dichlorophenol, chlorophenols, etc. These phenolic compounds are bioaccumulative in nature (air, water, food, animals, and plants), and due to their persistence in nature and their high toxicity it is imperative that they and their derivatives be detected quickly via in situ monitoring.

Compact systems suitable for on-site measurements of phenols are preferred, since they offer the option of rapid warning and avoid the errors and delays inherent in laboratory-based analyses [168]. The optical microfluidic detection methods presented in Table 5 can detect phenolic compounds such as phenol, bisphenol A (BPA), dopamine, [102], and catechol by fluorescence (LR: $9.79 \cdot 10^{-6}$ to $7.50 \cdot 10^{-4}$ M) [169] or colorimetric detection (LOD: 2 μM, LR: 5–70 μM [170]. Table 6 shows a summary of electrochemical microfluidic detection methods for phenols or phenolic compounds, device substrate and fabrication methods used for microchips and electrodes.

**Table 5.** Optical microfluidic detection methods for phenols or phenolic compounds.

| Samples | Device Substrate (or Components) | Detection Method (and/or Mechanism) | Fabrication Method | Analyte (Target) | Limit of Detection (LOD) Linear Range (LR) | Ref. |
|---|---|---|---|---|---|---|
| Tap water and river water samples | Fisher brand filter paper (P5; 09−801C) with a diameter of 11 cm and a medium porosity | Colorimetry | Inkjet printing and a layer-by-layer (LbL) assembly approach (formed by alternatively depositing layers of chitosan and alginate polyelectrolytes) onto filter paper | Phenolic compounds (phenol, bisphenol A (BPA), dopamine) | LOD: 0.86 (±0.1) μg/L | [102] |
| Environmental samples | Polyacrylamide film | Florescence (molecular absorption) | — | Catechol | LR: $9.79 \times 10^{-6}$ to $7.50 \times 10^{-4}$M | [169] |
| -Standard solutions (mixtures) of catecholamines; -Human urine and plasma samples | Fused silica fiber coated with a polystyrene/ divinylbenzene resin (PS/DVB) film | Optical fiber biosensor + chromatographic separation | Mechanically uncladded; enzymatic cladding; dip-coating of single optical fibers (OFs) | Dopamine, norepinephrine, epinephrine | LOD for dopamine: 2.1 pg/mL; LOD for norepinephrine: 2.6 pg/mL; LOD for epinephrine: 3.4 pg/mL | [171] |

**Table 5.** *Cont.*

| Samples | Device Substrate (or Components) | Detection Method (and/or Mechanism) | Fabrication Method | Analyte (Target) | Limit of Detection (LOD) Linear Range (LR) | Ref. |
|---|---|---|---|---|---|---|
| Homogeneous stock sol–gel solution | Hybrid Nafion/sol–gel silicate glass | Optical biosensors (crosslinking immobilization method of laccase and 3-methyl2-benzothiazolinone hydrazone (MBTH) | MBTH mixture was deposited onto a glass slide and coated | Catechol | LOD: 0.33 mM LR: 0.5–8.0 mM | [172] |
| Catechol in water sample | $Fe_3O_4$@Au core–shell nanoparticles | Colorimetric detection (absorbance principle) | Laccase-Au-$Fe_3O_4$ nanoparticles (NPs) | Catechol | LOD: 2 μM LR: 5–70 μM | [170] |

Since most phenols are oxidizable at moderate potentials, amperometry can serve as a highly sensitive tool for their detection [168]. The amperometric tyrosinase (Tyr)-based biosensors constitute promising technology for in situ phenol monitoring in discrete or batch systems because of a number of advantages (i.e., high selectivity, easy automation, fast response, potential for miniaturization, simple instrumentation, and low production cost) compared to classic procedures, including instrumental methods.

Mayorga-Martinez et al. developed an amperometric $CaCO_3$-PEI/Tyr-based biosensor integrated in a flow microsystem, which is presented schematically in Figure 8b. The electrochemical microfluidic-integrated biosensor was composed of PDMS/glass, with a graphite WE. The microchannel was fabricated in PDMS by soft lithography, and screen-printed electrodes (SPEs) modified with $CO_3$-polyethyleneimine were used.

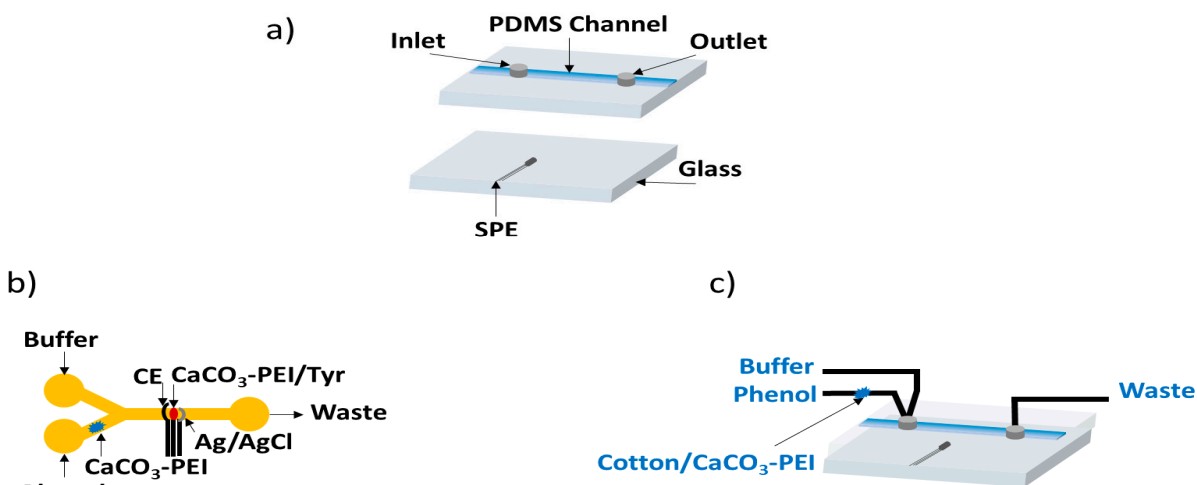

**Figure 8.** The construction of a fluidic microsystem for phenol sensing: (**a**) schematic representation of the PDMS/glass microchip device components; (**b**,**c**) schematic diagrams of the integrated dual microfluidic system for phenol removal and sensing (PEI—poly(ethyleneimine); Tyr—tyrosinase).

The $CaCO_3$-PEI/Tyr biosensor for phenol detection was evaluated by chronoamperometry. The biosensors showed a rapid and sensitive bioelectrocatalytic response, reaching about 95% of the steady-state current within 40s after each phenol-addition step. The obtained biosensing performance was LOD: 10 nM; LR: 0.5 to 5 μM [173]. The same microdevice (Figure 8c) was used for the detection of phenols via electrochemical impedance spectroscopy (EIS) [148]. They obtained good analytical performance in phenol detection in terms of reproducibility, selectivity, sensitivity, and limit of detection (LR: 0.01–10 μM and LOD: 4.64 nM).

**Table 6.** Electrochemical microfluidic detection methods for phenols or phenolic compounds.

| Samples | Device Substrate (or Components) | Working Electrode Type | Detection Method | Fabrication Method | Analyte (Target) | Limit of Detection (LOD) Linear Range (LR) | Ref. |
|---|---|---|---|---|---|---|---|
| Domestic water supplies; sample solution: 2,4-dichlorophenol (2,4-DCP) mixed with Folin–Ciocâlteu (FC) reagent | Plastic microfluidic chip with incorporated electrodes | Platinum | Potential difference measurements | Sputtering method of deposition of electrodes on plastic film | 2,4-Dichlorophenol | LOD: 0.1 ppm | [167] |
| Contaminated water sample with phenols | Hybrid PDMS/glass microchip | Graphite | Chronoamperometry. | Soft lithography in PDMS of microchannel; SPE modified with a $CO_3$-poly (ethyleneimine) (PEI) microparticles (MPs) and tyrosinase (Tyr) | Phenols | LOD: 10 nM LR: 0.5 to 5 µM | [173] |
| Contaminated water sample with phenols | Hybrid polydimethylsiloxane (PDMS)/glass chrono-impedimetric microchip; polyester substrate for screen-printed electrode (SPE) | Graphite | Electrochemical impedance spectroscopy (EIS); chrono-impedimetric detection of phenols | Soft lithography in PDMS of channels; sequential deposition of graphite ink and Ag/AgCl ink onto a glass substrate for a screen-printed electrode (SPE) | Phenols | LOD: 4.64 nM LR: 0.01–10 µM | [148] |
| Water samples | Polyethylene -based films | Carbon (screen-printed carbon electrodes) | Micellar electrokinetic chromatography with electrochemical detection (MEKC-EC); amperometric detector | Screen-printed carbon electrodes (SPCEs) | Trace phenolic compounds | LOD: $100 \times 10^{-12}$–$150 \times 10^{-12}$ M | [174] |
| Samplewaste; mixture of dopamine, epinephrine, catechol, and 4-aminophenol | Poly(dimethylsiloxane) (PDMS)silicon wafer | Cylindrical carbon electrodes | Cyclic voltammetry (CV) | Silicon wafer spin-coated with SU-8 2035-negative photoresistor; micromolding–casting process of liquid PDMS prepolymer | Dopamine, epinephrine, catechol, 4-aminophenol | LOD for dopamine: 140 nM; LR for dopamine: 140–45.00 µm LOD for epinephrine: 105 nM; LR for epinephrine: 0.105–47.90 µm LOD for catechol: 693 nM; LR for catechol: 0.693–188.10 µm LOD for 4-aminophenol: 459 nM LR for 4-aminophenol: 0.459–159.10 µm | [175] |
| Human blood and urine samples | Fiber optics; Teflon plug | Glassy carbon | Chromatography–electrochemical detector (HPLC-ED) | — | Epinephrine, dopamine, norepinephrine | LOD for epinephrine: 3.5 pg/mL LOD for dopamine: 2.9 pg/mL LOD for norepinephrine: 3.3 pg/mL LR: 5–125 pg/mL | [176] |

### 4.4.3. Microfluidic Detection Systems for Nitrites, Nitrates, and Ammonia

Environmental monitoring of nitrogen species—mainly nitrites and nitrates—is commonly performed using standard analytical techniques such as spectrophotometry, ion chromatography [146], laser-induced fluorescence (LIF), electrochemical detection (ED), and mass spectrometry (MS). For example, Fuji et al. [140] used a PDMS-based optical microfluidic chip for the simultaneous determination of sulfites and nitrites in aqueous samples (river-, pond-, and rainwater) by laser-induced fluorescence (LIF).

The schematic representation of the experimental setup of the integrated analytical system for the simultaneous fluorescence determination of sulfites and nitrites is presented in Figure 9a. Another innovative method was presented by Lopez-Ruiz et al. [177], who developed a low-cost paper-based microfluidic device with a smartphone application for the measurement of nitrite concentrations based on image analysis. The application studied the change in the hue (H) and saturation (S) coordinates of the HSV color space for different sensing areas by using a customized algorithm for the processing of an image taken with the built-in camera. The results (LOD 0.52 mg/L) show good use of a mobile phone as an analytical instrument [177]. In Table 7, a few optical microfluidic detection methods for nitrites and nitrates are presented, along with certain characteristics of the device used, fabrication methods, and the performance of each microfluidic device.

**Table 7.** Optical microfluidic detection methods for nitrites and nitrates.

| Samples | Device Substrate (or Components) | Detection Method (and/or Mechanism) | Fabrication Method | Analyte (Target) | Limit of Detection (LOD) Linear Range (LR) | Ref. |
|---|---|---|---|---|---|---|
| Aqueous samples (river-, pond-, and rainwater) | PDMS/glass microchip | Laser-induced fluorescence (LIF) | Microchannels made by photolithography and wet-etching methods; microfabricated glass template | Nitrites | LOD: $0.4 \times 10^{-6}$ M | [140] |
| Drinking water containing nitrites | PMMA microfluidic chip | Colorimetric chemical analysis (Griess method for nitrite detection on a chip) | Microchip fabrication: micromilling and solvent–vapor bonding procedure | Nitrites | LOD: $14 \times 10^{-6}$ M | [178] |
| Synthetic and natural water samples; environmental and drinking water | Whatman filter paper grade 1 and 4 | Colorimetry | Inkjet printing method of electrode; patterning grade 1 and 4 filter paper (Whatman) | Nitrites and nitrates | LOD for nitrites: 1 μm LOD for nitrates: 19 μm | [179] |
| Water samples | Standard laboratory Whatman paper grade 1 | Colorimetry | Stamping technique of the paper-based microfluidic devices | nitrites | LOD: 0.52 mg/L | [177] |

For real-time electrochemical detection, Gallardo-Gonzalez et al. [180] used a microfluidic device that consisted of PDMS (obtained by soft lithography) and a fully integrated chemical sensing platform (with four working microelectrodes, two Ag/AgCl reference microelectrodes, one Pt auxiliary electrode, and one counter microelectrode). The construction of the abovementioned fluidic microsystem for the detection of ammonium is presented in Figure 10a,b.

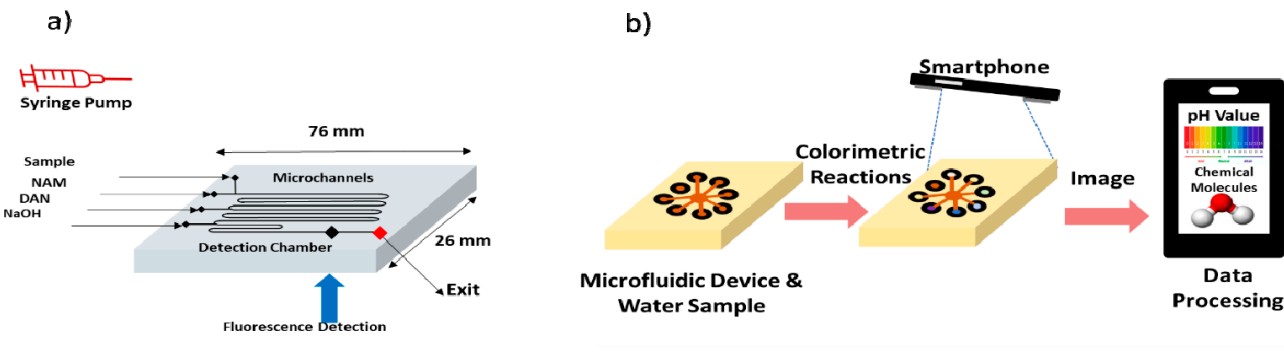

**Figure 9.** The construction of a fluidic microsystem for nitrites and sulfites: (**a**) schematic representation of the experimental setup of the integrated analytical system for the simultaneous fluorescence determination of sulfites and nitrites (**b**) schematic representation of a low-cost paper-based microfluidic device and smartphone application for the measurement of nitrite concentrations based on image analysis.

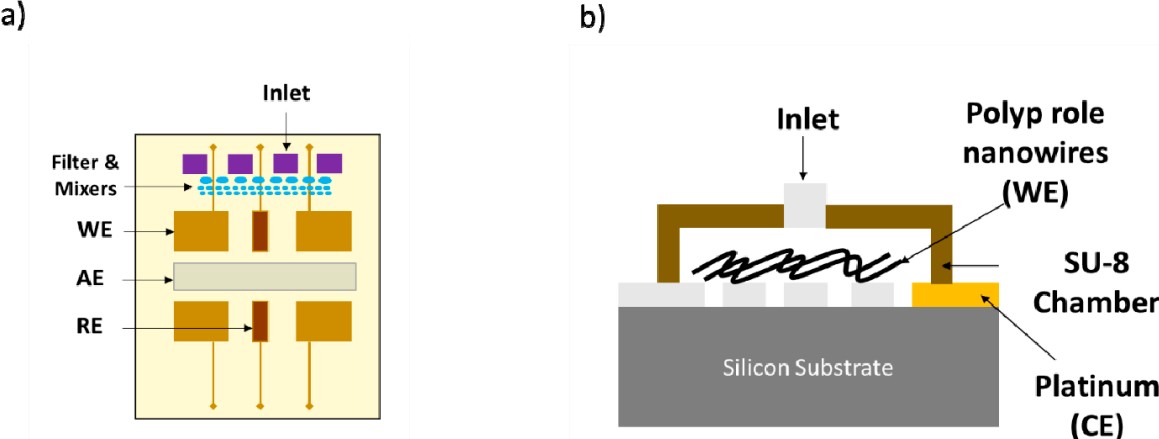

**Figure 10.** The construction of fluidic microsystems for the detection for ammonium and nitrate: (**a**) The components of transducers and negative-shaped silicon molds bearing the microfluidic elements. (**b**) The illustration of the electrochemical sensor chip.

The real-time potentiometric measurements in flowing water showed that the microfluidic device was still functional and responded to samples containing ammonium after being immersed in the sewage for at least 15 min. Therefore, the low-cost, low-power, easy-to-operate, miniaturized device developed by Gallardo-Gonzalez's team can be used for in situ and real-time potentiometric measurements in running water [180].

Aravamudhan et al. [181] developed a microfluidic nitrate-selective sensor based on polypyrrole-doped nanowires. Cyclic voltammetry, amperometry, and flow-through analysis were performed to evaluate the sensor's performance for the determination of nitrate ions in two sets of calibration solutions (DI water and IAPSO standard seawater). By using the electrochemical doping approach on polypyrrole nanowires, a highly sensitive (1.17–1.65 nA/µM) and selective nitrate sensor was demonstrated on an MEMS microfluidic platform. The sensor showed a linear response to nitrate of 10 µM (0.14 nitrate-N) to 1 mM (14 ppm nitrate-N) [181]. Table 8 shows synthetized electrochemical microfluidic detection methods for nitrites, nitrates, and ammonia.

**Table 8.** Electrochemical microfluidic detection methods for nitrites, nitrates, and ammonia.

| Samples | Device Substrate (or Components) | Working Electrode Type | Detection Method | Fabrication Method | Analyte (Target) | Limit of Detection (LOD) Linear Range (LR) | Ref. |
|---|---|---|---|---|---|---|---|
| Wastewater; ammonium-containing samples | PDMS microfluidic device; silicon substrate wafers | Gold | Cyclic voltammetry (CV) | Microelectrodes made by physical vapor deposition (PVD) followed by photolithography and lift-off; soft lithography and replica molding of PDMS microfluidic systems | Ammonium | LOD: $4 \times 10^{-5}$ M | [180] |
| Real-world samples; nitrate samples | Silicon substrate/polyimide protective layer | Silver thin film | Double-potential-step Chronocoulometry (DPSC) | Patterned polyimide insulation layer | $NO_3{}^-$ | LOD: 4–75 μM LR: 500–2000 μM | [182] |
| Seawater | Polypyrrole-covered carbon nanowire | Polypyrrole (PPy)-doped nanowires (NWs) on the interdigitated Pt | Double-potential-step chronocoulometry (DPSC) | Patterned electrochemical reagent chamber of the sensor chip using a thick SU-8 film; assembly of PPy NWs on the Pt lines using dielectrophoresis | $NO_3{}^-$ | LOD: 4.5 μM sensitivity: 1.17–1.65 nA/μM | [181] |
| Wastewater, tap water; river sample | Borosilicate glass tube | Carbon disk electrode modified with mesoporous carbon material (CMK-3) | Capillary electrophoresis with amperometric detection and electrochemical impedance spectroscopy | Carbon disk electrode constructed using a pencil lead | 1,3,5-Trinitro-benzene (TNB), 1,3-dinitrobenzene (DNB),2,4,6-trinitrotoluene (TNT), 2,4-dinitrotoluene (DNT) | LOD for TNB: 4 μg/L LOD for DNB: 4.1 μg/L LOD for TNT: 4.7 μg/L LOD for DNT: 3 μg/L LR for TNB: $10.7{-}4.7 \times 10^3$ μg/L LR for DNB: $8.4{-}3.7 \times 10^3$ μg/L LR for TNT: $11.4{-}5.0 \times 10^3$ μg/L LR for DNT: $9.1{-}4.0 \times 10^3$ μg/L | [183] |
| Dirty aquarium water samples (in the absence and presence of fishes) and Meia Ponte River water samples | Commercial glass substrate for device(borosilicate glass microchips with integrated electrodes) | Integrated electrodes | Capacitively coupled contactless conductivity detection (C4D) | —— | $NO_3{}^-$ $NO_2{}^-$ | LOD for $NO_3$: 4.4 μM LOD for $NO_2$: 4.9 μM | [146] |
| River water, tap water, mineral water | PMMA microchip, Isotachophoresis (ITP) and column-coupled capillary-zone electrophoresis (CZE) | Thin-film platinum electrodes | Conductivity | Microchip fabrication: substrate hot embossing; metallization of the PMMA covers used as the cover plates; sputtering deposition of thin-film platinum electrodes | Nitrites | LOD: 0.5–0.7 μM | [184] |

### 4.4.4. Microfluidic Detection Systems for Pathogens

Pathogens are infectious microorganisms such as bacteria, viruses, protozoans, fungi, or other microorganisms that can cause diseases in humans, animals, and plants. The most common pathogens with absorbance techniques are *Escherichia coli*, *Saccharomyces cerevisiae*, and *Aeromonas hydrophila* [136,185]. Many researchers have studied cholera toxins, *Bacillus globigii* [186], Staphylococcal enterotoxin B [187], *Listeria monocytogenes*, *Salmonella* [188], and *E. coli* [189] using fluorescent techniques. The parameters/performance of the optical microfluidic systems/biosensors, along with the components and fabrication methods of the devices, are presented in Table 9.

**Table 9.** Optical microfluidic detection methods for pathogens.

| Samples | Device Substrate (or Components) | Detection Method | Fabrication Method | Analyte (Target) | Limit of Detection (LOD) Linear Range (LR) | Ref. |
|---|---|---|---|---|---|---|
| Samples of microorganism-infected water | Glass substrate; dry-film resist (DFR)-basedmicrofluidic chip bonded with multimode fiber pigtails | Absorbance measurements(optical) | Photolithographic fabrication of microchannels | *Escherichia coli, Saccharomycescerevisiae,* and *Aeromonas hydrophila* | LOD for *A. hydrophila* and *E. coli*: $1.0 \times 10^5$ cells/mL LOD for *S. cerevisiae*: $1.0 \times 10^6$ cells/mL | [185] |
| Strains of Aeromonas hydrophila | Glass substrate | Absorbance measurements (optical) | Photolithographic fabrication of microchannels | *Aeromonas hydrophila* | LOD: 6 μL or 102 cells/mL | [136] |
| Infected water samples | Soda lime glass substrate of microfluidic chip (NS-12A, PerkinElmer, USA) | Fluorescence detection | - | *E. coli* | LOD: $10^4$ CFU/mL | [189] |
| Real samples; biological samples; spiked drinking water | Glass fiber; nitrocellulose membrane; integrated paper-based biosensor; hydrophobic PVC layers; separation of paper | Lateral flow assays (LFA) for bacterial nucleic acid detection; colorimetry | Cell deposition | *E. coli* | LOD: 10 CFU/mL (Water) | [190] |
| Samples containing mixtures of analytes | PDMS/glass | Fluorescence | - | *Cholera toxin; Staphylococcal enterotoxin B; Bacillus globigii* | LOD for *Cholera toxin*: 8 ng/mL; LOD for *Staphylococcal enterotoxin B*: 4 ng/mL; LOD for *Bacillus globigii*: $6.2 \times 10^4$ cfu/mL | [186] |
| Phosphate-buffered saline samples | Polyethylene channel | Fluorescence | - | *Staphylococcal enterotoxin B* | LOD: 5 ng/mL | [187] |
| Chicken carcass wash samples | Glass/hybrid | Fluorescence | - | *E. coli* | LOD: 20 organism | [191] |
| Real samples | 3D PDMS sponge | Fluorescence | The powdered salt particles were rubbed by adding water and then cast into molds (empty syringe) to shape the template for a PDMS sponge | *Listeria monocytogenes, Salmonella* sp. *Salmonella typhimurium* | LOD for: $10^3$ to $10^4$ CFU/mL LOD for: $1.5 \times 10^2$ CFU/ mL | [188] |

Compared with the traditional approaches, various electrochemical biosensors have been also constructed and used to detect pathogens, due to their advantages of simplicity, low cost, sensitivity, and easy miniaturization [192,193]. The principle of electrochemical biosensors for pathogens is mainly based on the specific recognition between various identification elements and targets, which can lead to changes in the detectable signal. For instance, Liu and coworkers [194] fabricated a facile, label-free, cheap electrochemical

Salmonella biosensor with satisfactory performance. The sensor also showed its specificity among different Salmonella serotypes, selectivity for different types of bacterial cells, and ability to distinguish between dead and live cells with a total detection time of 1 hour. The characteristics and construction of these biosensors can be found in Figure 11a,b.

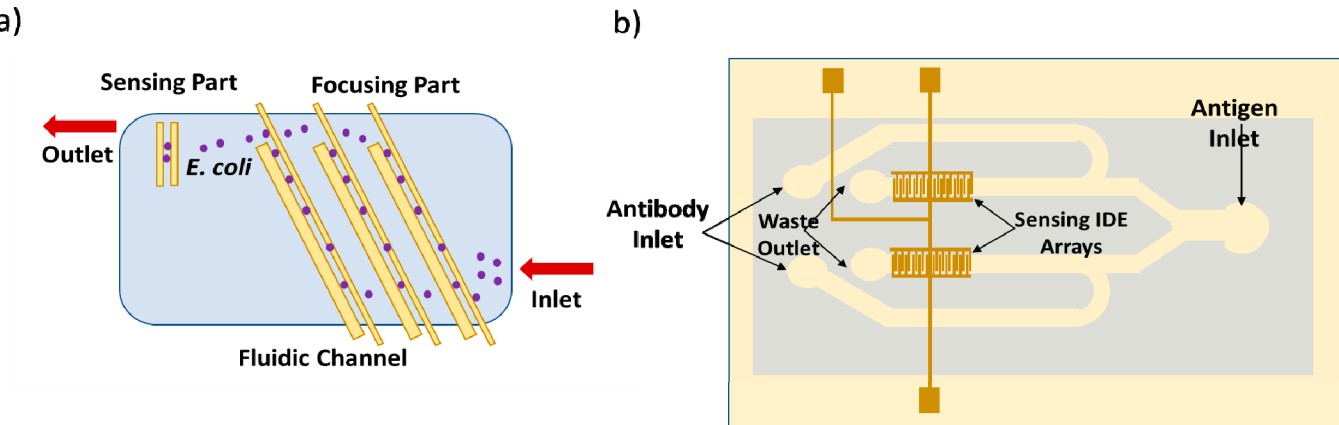

**Figure 11.** The construction of electrochemical microfluidic systems for the detection of pathogens: (**a**) a microfluidic sensor with a region that employs dielectrophoretic impedance measurements for the detection of *E. Coli* microorganism; (**b**) the schematic representation of a microfluidic biosensor for *Salmonella*.

Moreover, a microfluidic device for label-free detection of *Escherichia coli* in drinking water was developed by Myounggon et al. [195]. This type of microfluidic sensor, as shown in Figure 11a,b, can accurately quantify microorganisms that are present in low numbers (<100 CFU/mL) in a high-throughput manner.

In Table 10, the electrochemical microfluidic detection methods for *E. coli*, *S. aureus*, *Salmonella serogroups*, etc., along with the characteristics of the microfluidic devices and their performance, are summarized.

The information presented in this chapter is summarized in Figure 12. In this figure, (i) the different sample types used in pollution analysis and monitoring of the contaminants, (ii) the device substrate types, and (iii) the materials of the working electrode used as sensing units for electrochemical sensors integrated into the microfluidic devices are shown schematically, along with (iv) the detection methods for both types of microfluidic sensors.

**Table 10.** Electrochemical microfluidic detection methods for pathogens.

| Samples | Device Substrate (or Components) | Working Electrode Type | Detection Method (and/or Mechanism) | Fabrication Method | Analyte (Target) | Limit of Detection (LOD) Linear Range (LR) | Ref. |
|---|---|---|---|---|---|---|---|
| Bacteria-contaminated drinking water samples; mixture of bacterial suspensions | PDMS microfluidic chip | Gold | Dielectrophoretic impedance measurements | Conventional photolithographic and soft lithographic techniques for a PDMS microfluidic chip; PVD (sputtering) for the electrode material | *E. coli* | LOD: 300 CFU/mL | [195] |
| Mixed bacterial sample of *E. coli* O157:H7 and *S. aureus* | Polyethylene glycol (PEG)-based microfluidic chip integrated with a functionalized nanoporous alumina membrane | Platinum | Linear sweep voltammetry (LSV) | Soft lithography techniques | *E. coli* and *S. aureus* | LOD: 100 CFU/mL | [196] |
| Real sample | Poly (dimethylsiloxane) (PDMS) substrate | Carbon | Linear sweep voltammetry (LSV) | Soft lithography techniques for microchannels | *E. coli (DNA)* | LOD: 24 CFU/mL | [197] |
| *E. coli* samples | Poly(methyl methacrylate) (PMMA)/silicon dioxide wafer | Gold | Cyclic voltammetry and amperometric measurements | - | *E. coli* | LOD: $1.99 \times 10^4$–$3.98 \times 10^9$ CFU/mL | [198] |
| Salmonella samples | PDMS/glass | Interdigitated electrode (IDE) | Impedance | Surface micromachining technology for sputtering of Cr and Au on top of glass (SU8 type); PDMS bonding to seal the microchannel | *Salmonella serogroups* | LOD: 7 cells/mL | [199] |
| Bacterial samples | Glass substrate | Interdigitated array and gold microelectrode | Impedance | 3D printing and PDMS casting of microchannels | *Escherichia coli O157:H7* | LOD: 12 CFU/mL | [200] |
| Salmonella-specific aptamer probes | SU-8 substrate and suspended carbon nanowire | Carbon nanowire electrodes | Electrical detection/chemiresistive | Nanowires were deposited by electrospinning; photolithography for SU-8 support structure. | *Salmonella typhimurium* | LOD: 10 CFU/mL | [201] |
| Real samples of *S. typhimurium* cells | PDMS/glass for substrate; graphene oxide (GO) nanosheets wrapped in carboxylated multiwalled carbon nanotubes (cMWCNTs) composite | GO-cMWCNTs microelectrode | Electrochemical detection | Soft lithography for PDMS microchannels; wet chemical etching process for fabrication of microelectrodes | *Salmonella typhimurium bacterial cells* | LOD: 0.376 CFU/mL | [202] |
| Listeria cells, magnetic nanoparticles (MNPs) modified with anti-Listeria monoclonal antibodies, and gold nanoparticles (AuNPs) modified with anti-Listeria polyclonal antibodies and urease | PDMS/glass | Interdigitated microelectrode | Impedance | 3D printing and surface plasma bonding | *Listeria monocytogenes* | LOD: $10^6$ CFU/mL | [203] |

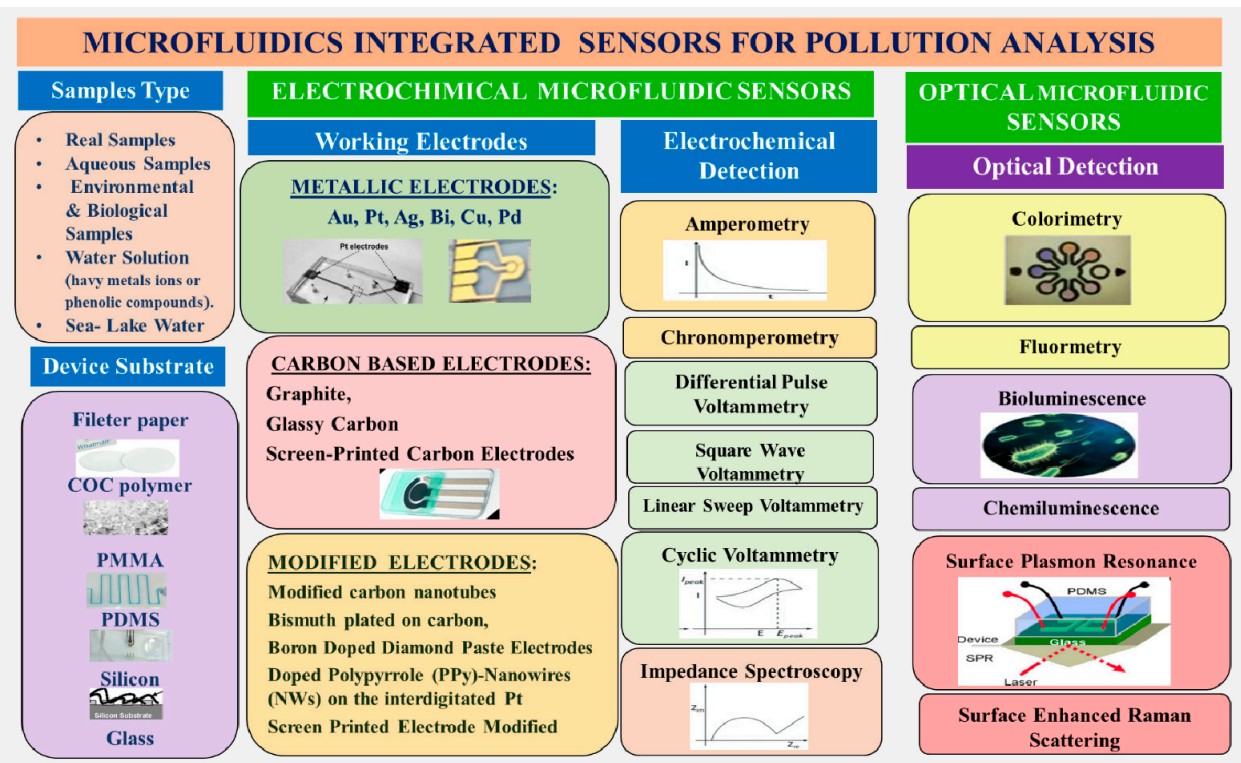

**Figure 12.** The schematic diagram of the two types of microfluidic sensors for pollution analysis, with sensing and detection units.

### 5. Conclusions and Future Perspectives

In line with the Goal 3 of the 2030 Agenda for Sustainable Development "Transforming our world: the 2030 Agenda for Sustainable Development"—"Ensure healthy lives and promote well-being for all at all ages"—and also consistent with the statement that "human well-being is closely linked to environmental health", it is necessary and beneficial for sustainable development that people have access to clean air to breathe, fresh water to drink, and places to live free of toxic substances and hazards. In this context, to support these vital needs, our overview of previous and recent research in the design and fabrication of optical and electrochemical microfluidic devices and microfluidics-integrated (bio)sensors for pollution analysis, in correlation with their environmental applications, offers a wide-ranging contribution to a synthetic picture of the most-used and best-performing microfluidic devices and their roles in field-monitoring measurements at lower cost and reduced pollutant reagent consumption.

In addition, the advantages and disadvantages of the various materials and techniques used for component fabrication, along with the benefits of miniaturization and integration of optical and electrochemical (bio)sensors in pollution analysis, were highlighted. Challenges in biosensors point to the need for the development of innovative portable analytical instruments that integrate optical or electrochemical sensors on microfluidic platforms. In the field of biosensors, further research and innovation should enable the manufacturing of sensitive and inexpensive portable microfluidic biosensors capable of monitoring soil contaminants, prompting timely action to prevent the spread of pollutants and contaminating agents in the environment. The availability of such integrated microfluidic biosensors could significantly reduce environmental pollution and enable continuous and real-time monitoring of environmental contaminants.

Future challenges consist of finding innovative ways to improve the reproducibility and reliability of microfluidic devices integrated into sensors, to increase their accuracy in detecting multiple contaminants simultaneously in the field. In the future, it is expected that the applicability of sensors integrated into microfluidic systems and other types of

microfluidic devices—for example, in the analysis of microplastic [204] or nanoplastic materials in rivers, lakes, or oceans—will be expanded.

**Author Contributions:** Conceptualization, B.A. and I.N.P.; methodology, B.A., I.N.P., and R.V.; validation, R.V. and I.N.P.; investigation, I.N.P.; writing—original draft preparation, I.N.P. and R.V.; writing—review and editing, B.A., I.N.P., R.V. and C.O.R. visualization, C.O.R.; supervision, R.V. All authors have read and agreed to the published version of the manuscript.

**Funding:** This research received no external funding.

**Informed Consent Statement:** Not applicable.

**Conflicts of Interest:** The authors declare no conflict of interest.

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
