# Peer review of "Microfluidic Devices and Microfluidics-Integrated Electrochemical and Optical (Bio)Sensors for Pollution Analysis: A Review"

_sustainability, doi:10.3390/su141912844_

Round 1
Reviewer 1 Report
This paper presents a thourough review of microfluidic devices for pollution analysis, covering the topics of materials, manufacturing processes and analysis techiques. This comprehensive review contains up-to-date information that will be of high interest to the scientific community.
Therefore, the paper is highly recommended for publication.
Next, we provide some minor suggestions.
- In most of the tables, the text of each row mixes with that of the contiguous rows. Therefore, in order to improve the clarity of the tables, it would be very convenient to separate the information of each row, for instance, with black lines or a blank rows.
- L 51. Is "labs-o chip" correctly spelled?
- L 56. "microfluidic deviceS"
-L 86. Spell correctly "micropollutants"
- L 154. ", ,"
- L 259. Is "...possesses can..." correct?
- L 337. There is a missing comma between "diamond" and "etc". The abbreviation "etc." is writtenn with a dot at the end. In a review paper, I wonder if "etc." should be replaced by the whole list of elements of the list or just removed
- L 562. Is "colourimetic" spelled correctly?
Author Response
Reviewers 1
We thank the reviewer for his/her thoughtful comments on the original version of the manuscript. We have revised the manuscript accordingly, making the required changes and additions that are highlighted in yellow in the resubmitted manuscript. In the pages below, we will respond in a point-by-point fashion to the reviewer’s comments. We agree with the reviewer’s comments which helped improve the clarity of our paper.
This paper presents a thorough review of microfluidic devices for pollution analysis, covering the topics of materials, manufacturing processes, and analysis techniques. This comprehensive review contains up-to-date information that will be of high interest to the scientific community. Therefore, the paper is highly recommended for publication.
Next, we provide some minor suggestions.
- In most of the tables, the text of each row mixes with that of the contiguous rows. Therefore, in order to improve the clarity of the tables, it would be very convenient to separate the information of each row, for instance, with black lines or blank rows.
R: We agree with the reviewer’s comment regarding the tables and we separated the information of each row with black lines, for all tables. Further, the publisher-editor will help format the tables if needed.
- L 51. Is "labs-o chip" correctly spelled?
R: We corrected the spelling errors, with “lab-on-chips”.
- L 56. "microfluidic deviceS"
R: We corrected the “microfluidic devices” according to the reviewer’s suggestion.
-L 86. Spell correctly "micropollutants"
- We corrected the spelling errors, and replaced with “micropolluants”.
- L 154. ", ,"
- We deleted one of the commas.
- L 259. Is "...possesses can..." correct?
- We deleted the word “possesses”.
- L 337. There is a missing comma between "diamond" and "etc". The abbreviation "etc." is written with a dot at the end. In a review paper, I wonder if "etc." should be replaced by the whole list of elements of the list or just removed
R: We agree with the reviewer’s comment and added the missing comma between "diamond" and "etc".
- L 562. Is "colourimetic" spelled correctly?
R: We corrected the spelling errors, with “colorimetric”

Reviewer 2 Report
This manuscript attempts to review the use of microfluidic electrochemical and optical biosensors for the evaluation of pollutants. Whilst a review that focussed on practical technologies for measuring, monitoring and evaluating the impact of pollutants and their implementation would be within the scope of a journal devoted to sustainability, this review does not achieve such an aim. The review has the following major deficiencies:
1. The link between the content of the review and the focus area of the journal is quite tenuous. The authors have not convincingly demonstrated how the technologies outlined in the review would make a substantive and immediate contribution to sustainability and how they might be used in a practical way to improve environmental outcomes and help to make human activities more sustainable. As such, I do not believe this review is suitable for publication in this journal.
2. The review is far too broad in scope. There are very many papers and reviews covering biosensors for environmental pollutants that utilise microfluidics in some way. As such, I do not feel that the review is adding any fresh insight either through updating the field with a summary of new literature or focussing on a particular well-defined sub-discipline worthy of special attention.
3. The references cited are a cursory selection of examples of technologies without any real context for how they are particularly important to advancing the field of sustainability. Because there are so many references you could potentially chose from as the selected field is so broad, this leads to citing and summarising references at random rather than providing a clear picture of the state of the art or explaining how these examples provide clear and realisable benefits for sustainability.
4. The review offers no meaningful overarching insights into the field. The conclusions section is quite shallow and leaves the reader with no idea of how close this technology is to final realisation as a practical pollutant monitoring tool and how such technology could realistically achieve positive impacts for sustainability now or in the future.
5. The review contains a lot of technical details on sensor design and construction that are of very little or no interest to readers of a journal on sustainability. Furthermore, some of the details provided on sensor operation are already very well known to analytical chemists and biochemists and so are superfluous even to an analysis of sensor operation.
In addition to these main points, there are further even more specific comments given below relating to errors and inconsistencies in the manuscript details as written.
6. The manuscript needs to be edited to correct spelling and grammar (there are too many instances to list individually).
7. Page 2 lines 70-71 “As is known . . . and air.” Water, soil and air in and of themselves don’t cause degradation of the natural environment. This sentence needs to be rephrased.
8. Page 3 line 104 Change “polystirene” to “polystyrene” here and throughout the manuscript.
9. The tables are poorly laid out. Individual entries need to be separated with horizontal lines as at present one entry is running on into the next, e.g. ‘Silicon (or silicon based substrates)’ needs to be separated properly from ‘Glass (or glass based substrates)’. Also, some of the text formatting is very poor with some columns set too narrow, e.g. page 27.
10. Page 5 Entries vii and ix for PDMS repeat each other and need to be condensed into one point.
11. Page 6 “moisturoptical” is not a word. This passage needs to be rewritten.
12. In Table 1 a number of the material types have no drawbacks listed. Are you concluding that they have no drawbacks? Your position on this should be clear in the manuscript.
13. Page 7 lines 128-132 “The mechanical properties . . . is 74.8 GPa.” These reference values are well known already. As they are not accompanied with any interpretive discussion they can be deleted from the text.
14. Table 2. The ‘Photolithography or wax printing’ and ‘wax printing’ entries are poorly distinguished from each other and overlap; they should therefore be combined in some way.
15. Page 11 lines 212-228 “In comparison with . . . the conventional ones.” At first you assert that optical biosensors have best sensitivity (a highly debateable assertion) and then you say that electrochemical sensors have the ‘most promising advantages’ in comparison to other transduction methods, which statement directly contradicts your first statement. Clarify your meaning by rewriting the text.
16. Page 12 line 268 “Selectivity is the most critical feature of a biosensor . . .” This statement is quite debateable. There are applications in which absolute selectivity is not required or desirable, e.g. if the analyst wanted to detect a class of compounds. As such, the degree of selectivity should be matched to the desired analytical outcome.
17. Page 12 lines 277-278 “A key parameter . . . (LOD).” Limit of detection is not equivalent to analytical sensitivity and should not be confused with it.
18. Page 12 line 293 “. . . sensor fabrication requires a linear response.” I disagree. Even with a linear response being desirable, there are still examples of non-linear responses that can be useful provided they are consistent, repeatable and predictable.
19. Page 12 lines 268-296 “Selectivity . . . of the device.” This text is completely superfluous as for sustainability scientists it lacks direct relevance to the applicability and impact of the sensors and for sensor scientists it is very well-established and widely known (except for the errors noted above).
20. One of the critical issues affecting microfluidics used with environmental samples is biofouling and clogging of the fluidics and the sensor surface, yet this major issue is not addressed in this review despite it having a direct impact on the real-world usefulness of the sensors.
21. Page 15 lines 407-411 “For environmental monitoring . . . [43, 141].” You could provide more details of the detection performance by quoting LODs here.
22. Page 15 lines 451-453 “Contactless conductivity . . . electroosmotic flow [168, 169].” It is not clear from your description how contactless conductivity detection is achieved. Clarify this by adding more detail.
23. Page 18 LOD and LR details are missing for the metal ions detected in seawater.
24. Table 7. Nitrogen dioxide is neither nitrite nor nitrate and is therefore outside the stated scope of this table.
25. Page 36 lines 686-687 “. . . microfluidic biosensors . . . of the environment.” You have not provided any evidence to back up this claim. A sensor can detect pollution but cannot in and of itself reduce pollution. This is an example of how you have not made any real connection between sensor device capabilities and sustainability.
Author Response
Reviewers 2
We thank the reviewer for his/her thoughtful comments on the original version of the manuscript. We have revised the manuscript accordingly, making the required changes and additions that are highlighted in yellow in the resubmitted manuscript. In the pages below, we will respond in a point-by-point fashion to the reviewer’s comments. We agree with the reviewer’s comments which helped improve the clarity of our paper.
This manuscript attempts to review the use of microfluidic electrochemical and optical biosensors for the evaluation of pollutants. Whilst a review that focussed on practical technologies for measuring, monitoring, and evaluating the impact of pollutants and their implementation would be within the scope of a journal devoted to sustainability, this review does not achieve such an aim. The review has the following major deficiencies:
- The link between the content of the review and the focus area of the journal is quite tenuous. The authors have not convincingly demonstrated how the technologies outlined in the review would make a substantive and immediate contribution to sustainability and how they might be used in a practical way to improve environmental outcomes and help to make human activities more sustainable. As such, I do not believe this review is suitable for publication in this journal.
R1:Starting from the definition "Environmental sustainability is the responsibility to conserve natural resources and protect global ecosystems to support health and well-being, now and in the future" and taking into consideration that "Because so many decisions that impact the environment are not felt immediately", we believe that this review is suitable for publication in Sustainability. Also, it is imperious that before reducing the sources of pollution, they should be monitored on the ground, immediately, with performance devices and at a low-cost price.
(see the editorial https://sphera.com/glossary/what-is-environmental-sustainability/?fbclid=IwAR38DZvPWYkYNOvwhfDxMClF7tOHrynmjAy1xGk_QcMZIQLmGXbX6FrUVLo).
The content of the article is based, I quote, on Lines 35-42 and Lines 59-62: Lines 35-42 we wrote this: „ scientists have been working together to find effective solutions for monitoring and reducing pollution sources by developing advanced materials or exploiting micro/nanodevice fabrication and integration of various processes in clean technologies for environmental sustainability [8-12].
Lines 59-62 One of the important advantages of using advanced materials or/and technologies, such as the microfluidic device integrated in biosensors, is the continuous and real-time monitoring of environmental contaminants such as toxic heavy metal ions, organic contaminants (phenols/phenolic compounds, pesticides/insecticides), pathogenic microorganism or gas pollutants [22, 23] for a sustainable environment.
The topic of our manuscript can be found and has a well-defined place in other topics of Sustainability journals, such as in Special Issue "Application of Microfluidic Methodology for Sustainability" of the same journal (https://www.mdpi.com/ journal/sustainability/special_issues/Application_microfluidic_methodology_sustainability). Based on the aforementioned, we are convinced that this review „Microfluidic Devices and Microfluidics Integrated Electrochemical and Optical (Bio)Sensors for Pollution Analysis -A review „ is suitable for publication in Sustainability journal.
- The review is far too broad in scope. There are very many papers and reviews covering biosensors for environmental pollutants that utilize microfluidics in some way. As such, I do not feel that the review is adding any fresh insight either through updating the field with a summary of new literature or focussing on a particular well-defined sub-discipline worthy of special attention.
R2: More than 200 bibliographic references have been cited in this review, including a great number of works from the last three years, thus contributing to the continued and updated feeding information for the readers in both fields, as well as a synthesis of data from the literature in well-structured chapters ( for instance Chapter 4- Microfluidic Detection Systems and Microfluidics Integrated (Bio)Sensors for Pollution Analysis) and subchapters on environmental applications of microfluidic devices for detection of the most common pollutants.
- The references cited are a cursory selection of examples of technologies without any real context for how they are particularly important to advancing the field of sustainability. Because there are so many references you could potentially choose from as the selected field is so broad, this leads to citing and summarising references at random rather than providing a clear picture of the state of the art or explaining how these examples provide clear and realizable benefits for sustainability.
R3: In line with Goal 3 ("Ensure healthy lives and promote well-being for all at all ages ") of "Transforming our world: the 2030 Agenda for Sustainable Development " (https://sdgs.un.org/2030agenda), and also in line with the statement that "human well-being is closely linked to environmental health", it is necessary and beneficial for sustainable development that people have access to clean air to breathe, fresh water to drink and places to live free of toxic substances and hazards. In this context, to support these vital needs, the development of microfluidic devices for monitoring environmental contaminants and the presentation of the contributions of scientists in the field of microfluidic devices and Microfluidics Integrated Sensors for Pollution Analysis, contribute to a systematic view of the most used and performing microfluidic devices (see Tables 1 and 2), and their role in monitoring for field measurements at low cost and low reagent consumption of pollutants (Tables 3 to 10), which determines taking timely measures (decision-making much faster) in order to isolate and to prevent the spread of pollutants and contaminating agents in the environment.
- The review offers no meaningful overarching insights into the field. The conclusions section is quite shallow and leaves the reader with no idea of how close this technology is to final realization as a practical pollutant monitoring tool and how such technology could realistically achieve positive impacts for sustainability now or in the future.
R4: We agree with the reviewer’s comment „The conclusions section is quite shallow” and we modified it accordingly. In the conclusion section, starting for your idea to explain why it is necessary and beneficial for sustainable development to use microfluidic systems, we added information regarding our contribution and highlight the benefits of using Microfluidic Devices and Microfluidics Integrated Electrochemical and Optical (Bio)Sensors for Pollution Analysis for environmental sustainability.
- The review contains a lot of technical details on sensor design and construction that are of very little or no interest to readers of a journal on sustainability. Furthermore, some of the details provided on sensor operation are already very well known to analytical chemists and biochemists and so are superfluous even to an analysis of sensor operation.
R5. The technical details on sensor design were introduced to help sustainability readers who are not familiar with this information. Also, the review fits into the topic of the Special Issue "Novel Developments in Advanced Materials and Technologies for Sustainable Environment"
In addition to these main points, there are further even more specific comments given below relating to errors and inconsistencies in the manuscript details as written.
- The manuscript needs to be edited to correct spelling and grammar (there are too many instances to list individually).
R6: We agree with the reviewer’s comments and we corrected the spelling errors, such as: “lab-on-chips” (line 51), “microfluidic devices” (line 56), “micropollutants” ((line 86), “colorimetric” (line 562), we deleted “possesses” ((line 259), etc.
- Page 2 lines 70-71 “As is known . . . and air.” Water, soil, and air in and of themselves don’t cause degradation of the natural environment. This sentence needs to be rephrased.
R7: We agree with the reviewer’s comments and we rephrased the sentence: „As is known, there are three major sources of pollutants that cause the degradation of the natural environment, namely water, soil and air pollutants.
- Page 3 line 104 Change “polystirene” to “polystyrene” here and throughout the manuscript.
R8: We agree with the reviewer’s comments and we corrected the spelling errors, with “polystyrene”.
- The tables are poorly laid out. Individual entries need to be separated with horizontal lines as at present one entry is running on into the next, e.g. ‘Silicon (or silicon-based substrates)’ needs to be separated properly from ‘Glass (or glass based substrates)’. Also, some of the text formatting is very poor with some columns set too narrow, e.g. page 27.
R9: We agree with the reviewer’s comment regarding the tables and we separate the information of each row with blank lines, for all tables.
- Page 5 Entries vii and ix for PDMS repeat each other and need to be condensed into one point.
R10: We agree with the reviewer’s comment and we condensed (ix) into (vii) ; we deleted (ix) and renumber the characteristics.
- Page 6 “moisturoptical” is not a word. This passage needs to be rewritten.
R11: We agree with the reviewer’s comment and we deleted from the text the word „moisturoptical”.
- In Table 1 a number of the material types have no drawbacks listed. Are you concluding that they have no drawbacks? Your position on this should be clear in the manuscript.
R12: We agree with the reviewer’s comment and we added:
Drawbacks:The cost of PMMA substrate per unit area is high [1]
Drawback: Don’t have mechanical flexibility (the drawback was inserted in Al oxide-based material raw)
Drawbacks: low transparency in the visible and near-UV spectrum (the drawback was inserted in Polycarbonate material raw)
- Page 7 lines 128-132 “The mechanical properties . . . is 74.8 GPa.” These reference values are well known already. As they are not accompanied by any interpretive discussion they can be deleted from the text.
R13: We agree with the reviewer’s comment and we deleted the sentence: “The mechanical properties . . . is 74.8 GPa.”
- Table 2. The ‘Photolithography or wax printing’ and ‘wax printing’ entries are poorly distinguished from each other and overlap; they should therefore be combined in some way.
R14: We separate the information of each row, with black lines for better visibility of the words in the Table 2 .
- Page 11 lines 212-228 “In comparison with . . . the conventional ones.” At first, you assert that optical biosensors have the best sensitivity (a highly debatable assertion) and then you say that electrochemical sensors have the ‘most promising advantages’ in comparison to other transduction methods, which statement directly contradicts your first statement. Clarify your meaning by rewriting the text.
R15: We agree with the reviewer’s comment and we deleted the sentence „In comparison with other types of biosensors, the optical biosensor has the best sensitivity”
- Page 12 line 268 “Selectivity is the most critical feature of a biosensor . . .” This statement is quite debateable. There are applications in which absolute selectivity is not required or desirable, e.g. if the analyst wanted to detect a class of compounds. As such, the degree of selectivity should be matched to the desired analytical outcome.
R16: We agree with the reviewer’s comment and we added it in the text (e.g. in case of interaction of an antigen with the antibody).
- Page 12 lines 277-278 “A key parameter . . . (LOD).” The limit of detection is not equivalent to analytical sensitivity and should not be confused with it.
R 17: We agree that the limit of detection is not equivalent to analytical sensitivity and we deleted this sentence.
- Page 12 line 293 “. . . sensor fabrication requires a linear response.” I disagree. Even with a linear response being desirable, there are still examples of non-linear responses that can be useful provided they are consistent, repeatable and predictable.
R 18: We don’t agree with the reviewer’s comment. Obtaining linear or non-linear responses can be determined based on the objectives of the fabricated devices. Even though the observation of non-linear responses leads to consistent, repeatable, and predictable, from the instrumentation point of view, a linear response is highly desirable in the fabrication of the sensor.
- Page 12 lines 268-296 “Selectivity . . . of the device.” This text is completely superfluous as for sustainability scientists it lacks direct relevance to the applicability and impact of the sensors and for sensor scientists it is very well-established and widely known (except for the errors noted above).
R19. We agree with the reviewer’s comment, and we cut lines 277-279. Because the other reviewers were agree of the rest of the paragraphs, we let them with corrections and added according with points 17 and 18.
- One of the critical issues affecting microfluidics used with environmental samples is biofouling and clogging of the fluidics and the sensor surface, yet this major issue is not addressed in this review despite it having a direct impact on the real-world usefulness of the sensors.
R20: Many challenges have been addressed in many other review papers; however, this is out of the scope of this review. The review focuses on the most common and newest materials used in the fabrication of microfluidic devices with integrated sensors and biosensors.
- Page 15 lines 407-411 “For environmental monitoring . . . [43, 141].” You could provide more details of the detection performance by quoting LODs here.
R21. We agree with the reviewer’s suggestions and we included in the text more details of the detection performance including LODs: In their experiments [133], the integrated membrane/electrochemical sampling sensor pursued trace monitoring of uranium and nickel using propyl gallate (PG) and dimethylglyoxime (DMG) chelating agents. These tests established adsorptive stripping protocols for trace uranium and nickel based on complexation with PG and DMG. Experimental variables including reagent delivery rate and ligand concentration were used to characterize and test the experimental stripping probe. Despite internal dilution, the renewable-flow probe results in extremely low detection limits, such as: 0.9 μg/L (1.5×10-8 M) for nickel and 10 μg/L (4.2×10-8 M) for uranium.
- Page 15 lines 451-453 “Contactless conductivity . . . electroosmotic flow [168, 169].” It is not clear from your description how contactless conductivity detection is achieved. Clarify this by adding more detail.
R22. We agree with the reviewer’s suggestions and we included in the text more details: “Contactless conductivity is one the important technique to detect inorganic or small organic ions in electrophoresis. It is preferred due to the electrode fouling, bubble formation due to water electrolysis, and interference with high voltages used to drive electroosmotic flow [168, 169]. Conductivity detection can be achieved either by direct contact of the mixture with the sensing parts or by a contactless method where the sensing electrodes are not attached directly to the measured mixture. This process requires a detector cell that basic part of the electronic circuitry. To evaluate the performance of the contactless conductivity detection, two major issues need to be addressed such as the noise analysis and the detector’s sensitivity“.
- Page 18 LOD and LR details are missing for the metal ions detected in seawater.
R23. We agree with the reviewer’s observations and we include the LOD and LR details:
LOD for Fe (II): 27 nM
LOD for Mn (II): 28 nM
LR for Fe (II): 27 -200 nM
LR for Mn (II): 0.028 - 6 mM
- Table 7. Nitrogen dioxide is neither nitrite nor nitrate and is therefore outside the stated scope of this table.
R24. We agree with the reviewer’s observations and we deleted the entire row about Nitrogen dioxide from table 7.

Reviewer 3 Report
Ms title: Microfluidic Devices and Microfluidics Integrated Electrochemical and Optical (Bio)Sensors for Pollution Analysis- A review
Journal: Sustainability
Comments: The manuscript focuses on the review of microfluidic devices, and the sensors based on the microfluidic approach are nicely written by covering all the fundamental aspects of the sensors, and their analytical characteristics, Methodology in fabrication in detail. The tables presented in the manuscript also give full-depth information about the sensory characteristics. Therefore, I do not have any comments here, but I found many formatting errors in the manuscript and many short sentences. Thus, I recommend authors please check those short sentences and continue with the main paragraph. For example,
1. Line 51, labs-o chips. should lab on a chip
2. Line 130, SiO2. it should be subscript.
3. Line 183, references should add together.
4. Line 251, “Recently, miniaturized electrochemical biosensors have the advantage of real-time 251 monitoring and label-free detection of biomarkers” The single sentence for a paragraph, should be combined with added sentences for a proper paragraph.
5. Similarly, line 316. The resulting curve is called a cyclic voltammogram or CV.
6. Like this many short sentences should be combined to make into a reasonable paragraph by generating some meaningful explanations.
7. I would also encourage the authors to include some more relevant figures that support their text in the manuscript.
Author Response
Reviewers 3
We thank the reviewer for his/her thoughtful comments on the original version of the manuscript. We have revised the manuscript accordingly, making the required changes and additions that are highlighted in yellow in the resubmitted manuscript. In the pages below, we will respond in a point-by-point fashion to the reviewer’s comments. We agree with the reviewer’s comments which helped improve the clarity of our paper.
Comments: The manuscript focuses on the review of microfluidic devices, and the sensors based on the microfluidic approach are nicely written by covering all the fundamental aspects of the sensors, and their analytical characteristics, Methodology in fabrication in detail. The tables presented in the manuscript also give full-depth information about the sensory characteristics. Therefore, I do not have any comments here, but I found many formatting errors in the manuscript and many short sentences. Thus, I recommend authors please check those short sentences and continue with the main paragraph. For example,
- Line 51, labs-o chips. should lab on a chip
R1: We corrected the spelling errors, with “lab-on-chips”.
- Line 130, SiO2. it should be subscript.
R2: We made a correction of SiO2.
- Line 183, references should add together.
R3: R: We agree with the reviewer’s comment, and put references together.
- Line 251, “Recently, miniaturized electrochemical biosensors have the advantage of real-time 251 monitoring and label-free detection of biomarkers” The single sentence for a paragraph, should be combined with added sentences for a proper paragraph.
R4: We agree with the reviewer’s comment and the mentioned single sentence was combined with the previous sentences for a proper paragraph.
- Similarly, in line 316. The resulting curve is called a cyclic voltammogram or CV.
- Like this many short sentences should be combined to make into a reasonable paragraph by generating some meaningful explanations.
R5 and R6: We agree with the reviewer’s comment and the mentioned single sentence was combined with the previous sentences for a proper paragraph.
- I would also encourage the authors to include some more relevant figures that support their text in the manuscript.
R7. We agree with the reviewer’s suggestions and we included them at the end of chapter 4, Figure 12. Also, the resolution of all Figures has been improved, and Figures 5 to 11 were adapted from the original ones.

Round 2
Reviewer 2 Report
In response to my earlier review comments the authors have made some changes to the manuscript, mostly to correct some of the more specific and more narrowly-focussed concerns raised in my earlier review. However, it is my opinion that the manuscript has not fundamentally or substantively changed from the earlier version so my main concerns remain and as a consequence I cannot recommend acceptance of the manuscript. To summarise again, recapitulating and expanding on my earlier comments:
1. The link between the content of the review and the focus area of the journal is still tenuous. I still maintain that the authors have not convincingly demonstrated how the technologies outlined in their review would make a substantive and immediate contribution to sustainability; a clear link needs to be drawn between the technology and how its use can enhance sustainability practices, i.e. beyond just saying it monitors pollutants. More specifically, details are needed on how they might be used in a practical way to improve environmental outcomes and help to make human activities more sustainable. As such, I do not believe this review is suitable for publication in this particular journal. This opinion applies irrespective of whether some content on microfluidics has been previously featured in the journal as the key is how the technology achieves impact in the field of sustainability.
2. The review is far too broad in scope and this review is not differentiated from other previous offerings in this area including both other reviews and the very many papers and reviews covering biosensors for environmental pollutants that utilise microfluidics in some way. As stated earlier, I do not therefore feel that the review is adding fresh insight either through updating the field with a summary of new literature or focussing on a particular well-defined aspect of this field worthy of special attention.
3. The references cited are lists of examples of technologies without practical context for how they are specifically important to advancing the field of sustainability. I still feel that the review is citing and summarising references at random rather than providing a clear picture of the state of the art or explaining how these examples provide clear and realisable benefits for sustainability. It is particularly in the interpretation of the potential impact of the technology for sustainability management practices where I feel that the review is quite weak.
4. The review offers little in the way of meaningful overarching insights into the field. The conclusions section whilst improved from the earlier version still leaves the reader with the impression that the technology is distant from a final realisation as a practical pollutant monitoring tool and so it becomes questionable how such technology could realistically achieve positive impacts for sustainability now or in the near future.
5. The review contains a lot of abstruse details on sensor design and construction that I still feel are of little or no interest to the core readership of a journal on sustainability. Furthermore, some of the details provided on sensor operation are already very well known to analytical chemists and biochemists and so are superfluous even to an analysis of sensor operation. The authors have chosen to retain this detail on the grounds of explaining sensor operation to non-sensor scientists but there is still a lot of highly technical sensor materials and design detail with no such layperson’s explanation and critical links are still missing to practical in-field usage of the technology and integration into sustainability policies and procedures.
Amongst the more specific points there are also a number that remain unaddressed or inadequately addressed, namely:
6. The manuscript still needs to be thoroughly edited to correct spelling and grammar (there are still too many instances to list individually).
14. Table 2. The ‘Photolithography or wax printing’ and ‘wax printing’ entries are poorly distinguished from each other and overlap; they should therefore be combined in some way. The entries remain listed separately and so appear to be repeating each other. What is the difference between a category the covers “photolithography or wax printing” and one that covers “wax printing”. I still maintain the two should be combined in one entry.
16 & 18. Whilst I still disagree with aspects of the authors’ response on these two points, they have at least added more detail to refine their statements further, so the text as currently written is probably minimally acceptable.
19. Page 12 lines 264-295 “Selectivity . . . of the device.” I still feel that this text is completely superfluous. As stated before, for sustainability scientists it lacks direct relevance to the applicability and impact of the sensors and for sensor scientists it is very well-established and widely known (except for the errors noted previously). The authors state this has been included to aid non-sensor scientists but there is so much other highly technical detail provided with no background explanation so this looks like a half measure. In reality, it is more likely that persons interested in sustainability would like to know how to use the technology and how it achieves impact for sustainability practice.
20. As stated before: “One of the critical issues affecting microfluidics used with environmental samples is biofouling and clogging of the fluidics and the sensor surface, yet this major issue is not addressed in this review despite it having a direct impact on the real-world usefulness of the sensors.” The authors assert that this issue is out-of-scope but I completely disagree, I would argue it is more in scope than abstruse discussions of sensor surface chemistry or sensor transduction as it directly impacts how useful the devices are as sustainability-enhancing tools.
22. Page 16 lines 457-465 “Contactless conductivity . . . detector’s sensitivity.” Despite changes made here, it is still not clear from your description how contactless conductivity detection is achieved. What is the definition here of a detector cell and how does it achieve genuine contactless sensing? This needs further clarification to be meaningful.